# Compositional Generative Multiphysics and Multi-component Simulation

## Abstract

Multiphysics simulation, which models the interactions between multiple physical processes, and multi-component simulation of complex structures are critical in fields like nuclear and aerospace engineering. Previous studies often rely on numerical solvers or machine learning-based surrogate models to solve or accelerate these simulations. However, multiphysics simulations typically require integrating multiple specialized solvers—each responsible for evolving a specific physical process—into a coupled program, which introduces significant development challenges. Furthermore, no universal algorithm exists for multi-component simulations, which adds to the complexity. Here we propose compositional Multiphysics and Multi-component Simulation with Diffusion models (MultiSimDiff) to overcome these challenges. During diffusion-based training, MultiSimDiff learns energy functions modeling the conditional probability of one physical process/component conditioned on other processes/components. In inference, MultiSimDiff generates coupled multiphysics solutions and multi-component structures by sampling from the joint probability distribution, achieved by composing the learned energy functions in a structured way. We test our method in three tasks. In the reaction-diffusion and nuclear thermal coupling problems, MultiSimDiff successfully predicts the coupling solution using decoupled data, while the surrogate model fails in the more complex second problem. For the thermal and mechanical analysis of the prismatic fuel element, MultiSimDiff trained for single component prediction accurately predicts a larger structure with 64 components, reducing the relative error by 40.3% compared to the surrogate model.

## 1 Introduction

Multiphysics simulation involves the concurrent modeling of multiple physical processes—such as heat conduction, fluid flow, and structural mechanics—within a single simulation framework to accurately capture the coupling effects between different physical processes. Similarly, multi-component simulation focuses on simulating complex structures composed of multiple similar components. Component is defined as: a repeatable basic unit that makes up a complete structure. For example, the reactor core typically consists of hundreds or thousands of fuel elements arranged in a square or hexagonal pattern. These simulations are essential across various scientific and engineering disciplines, including nuclear engineering (Ma et al., 2022; Chen et al., 2021), aerospace engineering (Candeo et al., 2011; Wang et al., 2023a), civil engineering (Sun et al., 2017; Meyer et al., 2022), and automotive industry (Ragone et al., 2021). Despite their significance, both multiphysics and multi-component simulations share a common challenge: while simulating individual components or physical processes is relatively straightforward, modeling the entire system with all its interactions is vastly more complex.

Numerous numerical algorithms have been developed for multiphysics simulation, which are broadly categorized into loose coupling and tight coupling (Hales et al., 2015). Loose coupling involves solving each physical process independently while treating the others as constant. Solutions for one physical process are iteratively transferred to related physical processes until convergence is achieved, often using techniques like operator splitting (MacNamara & Strang, 2016) and Picard iteration (Terlizzi & Kotlyar, 2022). Tight coupling, on the other hand, assembles equations of all physical processes into a large system, solving them simultaneously (Knoll & Keyes, 2004). While

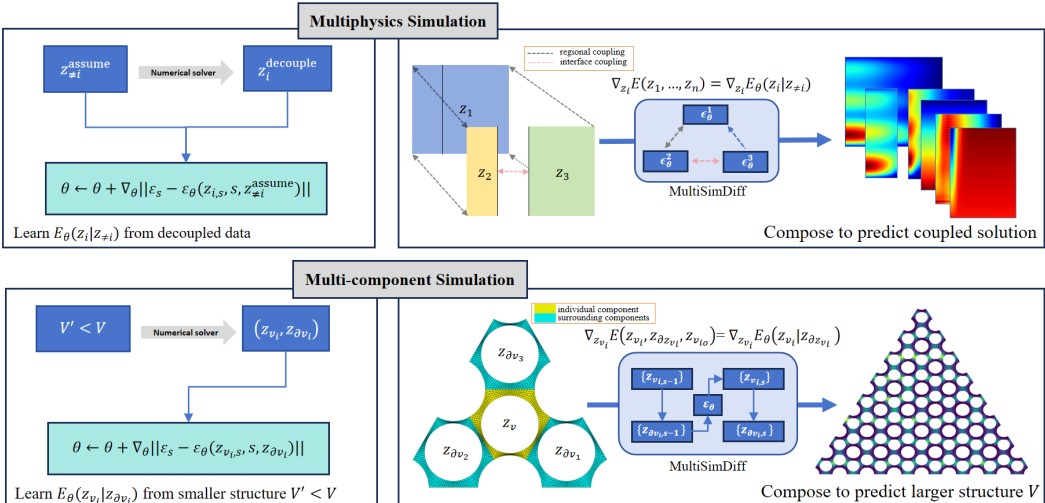

Figure 1: MultiSimDiff schematic. Our proposed algorithm can use models trained with decoupled data to predict coupled solutions (top) and use models trained with small structure simulation data to predict solutions for large structures (here 64 components)(bottom).

this method can potentially yield more accurate results, it encounters challenges such as high computational costs, varying spatial and temporal resolutions, and differing numerical methods across physical processes, leading to a more common use of loose coupling in engineering applications. In multi-component simulation, directly simulating the overall structure requires high computational cost and may encounter difficulties in convergence due to the increase in degrees of freedom. Substructure methods have been used in fields like nuclear engineering (Chen et al., 2021) and civil engineering (Sun et al., 2017) to reduce modeling and computational costs for repetitive components.

Despite advances in numerical algorithms, several significant challenges remain. In multiphysics simulations, considerable time and effort are required to develop programs that couple different specialized solvers. Furthermore, the complexity of the system increases due to coupling, requiring more computing resources. While some studies employ machine learning-based surrogate models to accelerate multiphysics simulations (Sobes et al., 2021; Park et al., 2021), these models still depend on coupled data for training, which necessitates the prior development of coupled numerical solution programs. In the case of multi-component simulations, the substructure method has primarily been applied to mechanical problems, with no widely applicable general method for multi-component systems. Consequently, current approaches often rely on selecting representative units or implementing simplifications for the analysis of complex structures, which can compromise the accuracy and scope of the simulation.

To address these challenges, we propose compositional Multiphysics and Multi-component Simulation with Diffusion models (MultiSimDiff). The core innovation of MultiSimDiff is its treatment of multiphysics and multi-component simulations as generative probabilistic modeling, where interactions between multiple physical processes or components are captured through composing learned energy functions conditioned on others in a structured way. In multiphysics simulation, MultiSimDiff generates coupled solutions (accounting for interactions between different physical processes) from decoupled data (assuming other fields are known and focused on solving a single field) by modeling the solutions of physical processes as a joint probability distribution. The solution for each individual process is treated as a conditional probability distribution, based on Bayes' theorem. By training diffusion models (Ho et al., 2020) on decoupled data, we capture these conditional distributions. During inference, the model combines these distributions and performs reverse diffusion to produce the coupled solution. For multi-component simulations, MultiSimDiff models each component's solution as a conditional probability distribution using the local Markov property, conditioned on neighboring components. By training diffusion models on small structures, we create conditional models for individual components. During inference, reverse diffusion is applied iteratively across all components, yielding the solution for the entire structure. We have mathematically

derived the principles why MultiSimDiff can obtain coupled solutions and large structure solutions in Sections 3.1 and 3.2. A schematic of MultiSimDiff is provided in Fig. 1.

We illustrate the promise of this approach through three challenging tasks. First, we demonstrate its capability for multiphysics simulation by applying it to coupled reaction-diffusion equations and nuclear thermal coupling combined with conjugate heat transfer. Second, we verify its capability in multi-component simulation through thermal and mechanical analysis of prismatic fuel elements.

Concretely, our contributions are threefold: (1) We introduce a novel approach, MultiSimDiff, for multiphysics and multi-component simulations, framing the problem in terms of joint probabilistic modeling. By training on decoupled (small structure) training data, MultiSimDiff can generate coupled (large structure) solutions. (2) We create and open-source benchmark datasets for both multiphysics and multi-component simulations, providing a valuable resource for future research. (3) Our method demonstrates success in both domains. For multiphysics simulation, MultiSimDiff accurately predicts coupled solutions in complex problems where surrogate models fail. In multi-component simulations, MultiSimDiff, trained on single components, accurately predicts larger structures with up to 64 components, reducing relative error by 40.3% compared to surrogate models.

## 2 RELATED WORK

**Multiphysics simulation.** Most existing studies develop unified surrogate models for all physical processes by coupling solutions(Tang et al., 2024; Ren et al., 2020; Park et al., 2021; Wang et al., 2023b). For complex problems, programs for each physical process are typically independent. It is often feasible to establish a surrogate model for one specific physical process and then integrate it with other numerical programs (El Haber et al., 2022; Han et al., 2019). Alternatively, surrogate models can be constructed separately for each physical processes and iteratively converged through an iterative process (Sobes et al., 2021). Because the purpose of our algorithm is to infer coupled solutions through models trained with decoupled data, and establishing the surrogate model for all physical processes requires coupling solution training models, we adopt the method of establishing surrogate models for each physical process separately as the baseline to validate the proposed algorithm.

**Multi-component simulation.** To our knowledge, there do not exist utilized machine learning methods specifically designed for multi-component simulation. A relevant study is the CoAE-MLSim algorithm (Ranade et al., 2021). This algorithm combines neural networks with numerical iteration. It first partitions the computational domain into multiple subdomains, and then trains a neural network to learn the flux conversation between subdomains. During inference, the neural network with flux conservation is applied sequentially in each subdomain, looping until convergence. We further extend this algorithm to multi-component simulation and use it as a baseline. Besides, graph neural network (GNN) (Wu et al., 2020) can learn on small graphs and inference on larger graphs (Xu et al., 2019); Graph Transformer (Kreuzer et al., 2021) employs the Laplacian matrix of a graph to characterize its structure, and by leveraging the Transformer architecture, it achieves learning on graphs. We also compare MultiSimDiff with GNN and Graph Transformer.

**Compositional models.** Recent research has extensively explored the compositional combination of generative models for various applications, including 2D image synthesis (Du et al., 2020; Liu et al., 2021; Nie et al., 2021; Liu et al., 2022; Wu et al., 2022; Du et al., 2023; Wang et al., 2023c), 3D synthesis (Po & Wetzstein, 2024), video synthesis (Yang et al., 2023a), multimodal perception (Li et al., 2022), trajectory planning (Du et al., 2019; Urain et al., 2023; Gkanatsios et al., 2024; Yang et al., 2023b), inverse design (Wu et al., 2024b), and hierarchical decision making (Ajay et al., 2024). A particularly effective approach for combining predictive distributions from local experts is the product of experts framework (Hinton, 2002; Cohen et al., 2020; Kant et al., 2024; Tautvaišas & Žilinskas, 2023). Their focus is on how a single object is influenced by multiple factors, such as generating images that meet various requirements in image generation (Du et al., 2023) or enhancing the lift-to-drag ratio under the influence of two wings in inverse design (Wu et al., 2024b). However, our problem involves multiple objects, such as multiple physical processes and components, requiring the capture of interactions between these fields or components. Existing research is not applicable to multiphysics and multi-component simulation. To the best of our knowledge, we are the first to introduce a compositional generative approach to multiphysics and multi-component sim-

ulations, demonstrating how this framework enables generalization to far more complex simulation tasks than those encountered during training.

## 3 METHOD

In this section, we introduce the principle of MultiSimDiff solving multiphysics and multi-component simulation in section 3.1 and section 3.2, respectively.

### 3.1 MULTIPHYSICS SIMULATION

Consider a complex multiphysics simulation problem that consists of multiple physical processes : $z = (z_1, z_2, \ldots, z_n)$, where each $z_i$ may contain one or more fields. For example, the mechanics contains the stress and strain fields in three directions. Each process $z_i$ has its own governing equation which depends on other processes , and solving equations for other processes also requires that process. Therefore, all equations must be solved *simultaneously* to achieve the most accurate representation of the physical system.

Simulating all the processes $z$ together can be challenging, while it will be simple if we simulate a single process $z_i$. By specifying the other processes $z_{\neq i} = (z_1, ..., z_{i-1}, z_{i+1}, ..., z_n)$ and the given outer inputs[1]

$$z_i = f(z_{\neq i}, C) \tag{1}$$

where $f$ is a numerical solver. Omitting the given condition $C$, then: $z_i = f(z_{\neq i})$. Now we consider the results of multiple physical processes as a joint probability distribution:

$$(z_1, z_2, ..., z_n) \sim p(z_1, z_2, ..., z_n) \tag{2}$$

For each process, we consider it as a conditional distribution: $z_i \sim p(z_i|z_{\neq i})$, which relates to the joint distribution via:

$$p(z_1, z_2, ..., z_n) = p(z_i|z_{\neq i})p(z_{\neq i}) \tag{3}$$

Writing the probability distribution in the form of (learnable) energy functions $E(z)$ (Du et al., 2023; LeCun et al., 2006), the energy functions relates to the joint probability of $z$, the conditional probability of $z_i$, and the marginal distribution of $z_{\neq i}$ respectively by:

$$\begin{cases} p(z) = \dfrac{1}{Z}e^{-E(z)} \\ p(z_i \mid z_{\neq i}) = \dfrac{1}{Z(z_{\neq i})}e^{-E(z_i|z_{\neq i})} \\ p(z_{\neq i}) = \dfrac{1}{Z_{\neq i}}e^{-E(z_{\neq i})} \end{cases} \tag{4}$$

where $Z, Z_{\neq i}$ are normalization coefficients (constants). Note that for $p(z_i|z_{\neq i})$, since $z_{\neq i}$ is the condition, the normalization $Z(z_{\neq i})$ depends on $z_{\neq i}$. Substituting Eq. 4 into Eq. 3, then taking logarithms of both sides, we have:

$$E(z) + \log Z = [E(z_i \mid z_{\neq i}) + \log Z(z_{\neq i})] + [E(z_{\neq i}) + \log Z_{\neq i}] \tag{5}$$

Taking the derivative w.r.t. $z_i$ on both sides, we have:

$$\nabla_{z_i} E(z_1, z_2, ..., z_n) = \nabla_{z_i} E(z_i|z_{\neq i}) \tag{6}$$

which uses the fact that $\log Z$, $\log Z_{\neq i}$, $\log Z(z_{\neq i})$, and $E(z_{\neq i})$ are all independent of $z_i$.

Eq. 6 is the foundation of our compositional multiphysics simulation method. We see that when sampling the joint distribution $p(z_1, z_2, ..., z_n)$, we can simply use the learned conditional diffusion model to sample each $z_i$, while using the estimated $z_{\neq i}^e$ of other physical processes as conditions. This means that to learn the multiphysics simulation of multiple physical processes $z_1, z_2, ..., z_n$, we no longer need to develop a coupled algorithm that simultaneously solves all physical processes. Instead, we can simply use decoupled solvers (each physical process is solved independently while

---

[1] In this paper, "outer inputs" refers to the inputs of the physical system.

---

**Algorithm 1** Algorithm for multiphysics simulation by MultiSimDiff.

---

**Require:** Compositional set of diffusion model $\epsilon_\theta^i(z_{i,s}, C, s), i = 1, 2, ..., N$, outer inputs $C$, diffusion step $S$, number of external loops $K$, number of physical processes $N$.

1: $z_i^e \sim \mathcal{N}(0, I)$ // initialize estimated fields $z_i^e$
   // add an external loop to improve the estimated fields $z_i^e$:
2: **for** $k = 1, ..., K$ **do**
3:     $\hat{z}_i^e \leftarrow z_i^e$ // update previous estimated fields $\hat{z}_i^e$
4:     $z_i^e \sim \mathcal{N}(0, \mathbf{I})$ // initialize current estimated fields $z_i^e$
5:     $z_{i,S} \sim \mathcal{N}(0, \mathbf{I})$ // initialize physical fields $z_i$
       // denoising cycle of diffusion model:
6:     **for** $s = S, ..., 1$ **do**
7:       $\lambda = 1 - \frac{s}{S}$ if $k > 1$ **else** 1 // define the weights of $\hat{z}_i^e$ and $z_i^e$
         // loops for each physical process:
8:       **for** $i = 1, ..., N$ **do**
9:         $w \sim \mathcal{N}(0, \mathbf{I})$
           // use weighted estimated fields as conditions for single step denoising:
10:      $z_{i,s-1} = \frac{1}{\sqrt{\alpha_s}}(z_{i,s} - \frac{1-\alpha_s}{\sqrt{1-\overline{\alpha}_s}}\epsilon_\theta^i(z_{i,s} \mid \lambda z_{\neq i}^e + (1-\lambda)\hat{z}_{\neq i}^e, C, s)) + \sigma_s w$
           // update the estimation of current field:
11:      $z_i^e = \frac{1}{\sqrt{\overline{\alpha}_s}}(z_{i,s} - \sqrt{1-\overline{\alpha}_s}\epsilon_\theta^i(z_{i,s} \mid \lambda z_{\neq i}^e + (1-\lambda)\hat{z}_{\neq i}^e, C, s))$
12:       **end for**
13:     **end for**
14: **end for**
15: **return** $z_{i,0}$

---

treating the other physical processes as known) to generate data, learn the conditional distributions $p(z_i|z_{\neq i}) \propto e^{-E(z_i|z_{\neq i})}$, and in the inference time, sample from the joint distribution via Eq. 6, achieving multiphysics simulation. During training, the energy $E(z_i|z_{\neq i})$ is implicitly learned via the diffusion objective below, which learns the gradient of the energy:

$$L_{\text{MSE}} = ||\epsilon - \epsilon_\theta(\sqrt{1-\beta_s}z_i + \sqrt{\beta_s}\epsilon; z_{\neq i}, s)||_2^2, \ \epsilon \sim \mathcal{N}(0, I) \tag{7}$$

where the denoising network $\epsilon_\theta(\cdot)$ corresponds to the gradient of the energy function $\nabla_z E_\theta(\cdot)$ (Du et al., 2023). During inference, we sample from the joint distribution $p(z_1, z_2, ...z_n)$ via (Ho et al., 2020):

$$z_{i,s-1} = \frac{1}{\sqrt{\alpha_s}}\left(z_{i,s} - \frac{1-\alpha_s}{\sqrt{1-\overline{\alpha}_s}}\epsilon_\theta^i(z_{i,s} \mid z_{\neq i}^e, s)\right) + \sigma_s w, \ w \sim \mathcal{N}(0, I) \tag{8}$$

$$z_i^e = \frac{1}{\sqrt{\overline{\alpha}_s}}\left(z_{i,s} - \sqrt{1-\overline{\alpha}_s}\epsilon_\theta^i(z_{i,s} \mid z_{\neq i}^e, s)\right) \tag{9}$$

for $s = S, S-1, ...1$ and $i = 1, 2, ...n$. Here, $z_i^e$ represents the estimated value for the $i$th field $z_i$, $z_{\neq i}^e = (z_1^e, ...z_{i-1}^e, z_{i-1}^e, ...z_n^e)$, and $\sigma_s$ is the noise level.

This iterative method is similar to the Expectation-Maximization (EM) algorithm (Moon, 1996), refining each variable's estimation based on current estimates of others. An external loop can be added to repeat the diffusion model's inference, using the previous step's physical fields to improve the initial estimate. The ablation study about hyperparameters $K, \lambda$ and the estimation method are discussed in Appendix F. The algorithm is shown in Algorithm 1. Line 2 is the external loop, while lines 6 to 11 represent the denoising cycle of the diffusion model. In each diffusion step, the physical physical processes are updated sequentially, with $z_{\neq i}$ using the estimated physical processes that have already been updated at this diffusion step.

### 3.2 MULTI-COMPONENT SIMULATION

Consider a complex structure that is composed of many components: $V = v_1 \cup v_2 \cup \ldots \cup v_n$, and the solution in each component $v_i$ is $z_{v_i}$. It should be noted that $v_i$ represents an entity here, and if there are multiple physical processes on this entity, it is also a multiphysics problem. Each component shares similarities and is arranged in a specific pattern, like an array, to compose this complex structure. Simulating the entire structure $V$ can be challenging while simulating an individual component

$v_i$ is easier. By specifying the boundary condition $z_{\partial v_i}$, the given outer inputs $C$, and the geometry $v_i$ of component $v_i$, we can compute $z_{v_i}$:

$$z_{v_i} = f(z_{\partial v_i}, C, v_i) \tag{10}$$

where $f$ is a numerical solver. The outer inputs $C$ and geometry $v_i$ are given conditions, $z_{\partial v_i}$ is boundary conditions. Omitting the given condition, then: $z_{v_i} = f(z_{\partial v_i})$. Then we divide the whole geometry $V$ to three parts: $V = v_i \cup \partial v_i \cup v_{io}$, where $v_{io}$ represents other parts of $V$ except $v_i \cup \partial v_i$. The solution of the whole geometry $V$ can be written as the following probability distribution:

$$(z_{v_1}, z_{v_2}, ..., z_{v_n}) = (z_{v_i}, z_{\partial v_i}, z_{v_{io}}) \sim p(z_{v_i}, z_{\partial v_i}, z_{v_{io}}) \tag{11}$$

Consider the complex structure as an undirected graph $G = (V, E)$, and the random variable $z_{v_i}$ is the property of component $v_i$. The graph $G$ satisfies the local Markov property: A variable is conditionally independent of all other variables given its neighbors. Thus, $z_{v_i}$ satisfies

$$(z_{v_i} \perp z_{V \setminus N[v_i]}) | z_{\partial v_i} \tag{12}$$

Here $\partial v_i$ is the set of neighbors of $v_i$, $N[v_i] = v_i \cup \partial v_i$, and $V \setminus N[v_i] = v_{io}$. By using this property of Markov random field, $p(z_{v_i}, z_{\partial v_i}, z_{v_{io}})$ can be written as:

$$p(z_{v_i}, z_{\partial v_i}, z_{v_{io}}) = p(z_i | z_{\partial v_i}) p(z_{v_{io}} | z_{\partial v_i}) p(z_{\partial v_i}) \tag{13}$$

Writing the probability distribution in the form of energy, and through the same derivation as in Section 3.1, we obtain:

$$\nabla_{z_{v_i}} E(z_{v_i}, z_{\partial v_i}, z_{v_{io}}) = \nabla_{z_{v_i}} E(z_{v_i} | z_{\partial v_i}) \tag{14}$$

Therefore, when sampling the joint distribution $p(z_{v_1}, z_{v_2}, ..., z_{v_n})$, we can simply use the learned conditional diffusion model to sample each $z_{v_i}$, while using the estimated $z_{\partial v_i}^e$ as conditions. The multi-component simulation can be achieved using an algorithm similar to multiphysics simulation. Since each $z_{v_i}$ is inferred with the same model, it can be processed together, improving inference efficiency by eliminating the need for additional loops for each physical process. We provide Alg. 2 in Appendix A for multi-component simulation. Additionally, we use the assumption of Markov random fields in the derivation. The rationality of this assumption and the application scenarios of the algorithm are discussed in Appendix J.

Our proposed framework of multiphysics simulation in Section 3.1 and multi-component simulation in Section 3.2, constitute our full method of compositional MultiPhysics and Multi-component Simulation with Diffusion models (MultiSimDiff). It circumvents the development of coupled programs that requires huge development efforts, and achieves multiphysics and multi-component simulation by composing the learned conditional energy functions according to the variable dependencies. Below, we test our method's capability in challenging engineering problems.

# 4 EXPERIMENTS

In the experiments, we aim to answer the following questions: (1) Can MultiSimDiff predict coupled solutions (accounting for interactions between different physical processes) from models trained in decoupled data (assuming other processes are known and focus on solving a single process)? (2) Can MultiSimDiff predict large structure solutions from a model trained in small structure data? (3) Can MultiSimDiff outperform surrogate model[2] in both tasks? To answer these questions, we conduct experiments to assess our algorithm's performance on two problems across three scenarios. In Section 4.1, we solve the reaction-diffusion equation. While it's not a classic multiphysics coupling issue since both quantities are part of concentration fields, we consider them as separate physical processes to validate the capability of MultiSimDiff for multiphysics simulation. Section 4.2 examines a more complex scenario involving various types of coupling: region, interface, strong, weak, unidirectional, and bidirectional to further test the algorithm's capacity to handle multiphysics simulation. Section 4.3 simplifies actual engineering problems to evaluate the algorithm's effectiveness with multi-component simulation. Each experiment uses two network architectures, training

---

[2]The surrogate model mentioned in this article involves inputting the system's input into a neural network to directly predict the output in one step, distinguishing it from the diffusion model, which uses the input as a condition to step-by-step generate the output through denoising.

both with their respective diffusion and surrogate models for comparison, employing consistent hyperparameters and settings to ensure fairness. The computational domains of experiment 1 and experiment 2 are on regular meshes, using Fourier neural operator (FNO) (Li et al., 2021; Lim et al., 2023) and U-Net (Ronneberger et al., 2015) as network architectures. The computational domain of experiment 3 is on a finite element mesh, using Geo-FNO (Li et al., 2023) and Transolver (Wu et al., 2024a) as the network architecture. Additionally, in experiment 3, we also compare MultiSimDiff with graph neural networks GIN (Xu et al., 2019) and Graph transformer SAN (Kreuzer et al., 2021). To highlight the difference between training and testing data, Appendix G calculates the Wasserstein distances between decoupled and coupled data in multiphysics problems, as well as between small and large structural data in multi-component problem, and visualizes them. The code is available at the anonymous repository. As another contribution to the community, we will also open-source the data to facilitate future method development of multiphysics and multi-component simulations. In Appendix K, we conduct a comparison of our dataset with existing scientific datasets (Takamoto et al., 2022).

## 4.1 REACTION-DIFFUSION

Reaction-diffusion (RD) equations have found wide applications in the analysis of pattern formation, including chemical reactions. This experiment uses the 1D FitzHugh-Nagumo reaction-diffusion equation (Rao et al., 2023), it has two concentration fields: $u, v$. The objective is to predict the system's evolution under different initial conditions. We use surrogate models that iteratively interact as a baseline. The training data consists of decoupled data, where other physical processes are assumed and treated as inputs to solve the equations governing the current physical process . In this experiment, a Gaussian random field (Bardeen et al., 1986) is employed to generate the other physical processes and initial conditions, and numerical algorithms are used to compute the solution of the current physical process. The validation data similarly consists of decoupled data not used during training. For the test data, initial conditions for both physical processes are generated using a Gaussian random field, and the ground-truth coupled solution is obtained using a fully coupled algorithm. Further details on the datasets, equation, network architecture, and training process are provided in Appendix B.

Table 1: Relative L2 norm of error on reaction-diffusion equation for multiphysics simulation.

| method | $u$ | | $v$ | |
| --- | --- | --- | --- | --- |
| | decoupled | coupled | decoupled | coupled |
| surrogate + FNO | 0.0669 | 0.0600 | 0.0080 | 0.0320 |
| **MultiSimDiff (ours)** + FNO | 0.0270 | **0.0290** | 0.0102 | **0.0264** |
| surrogate + U-Net | 0.0152 | 0.0184 | 0.0039 | **0.0174** |
| **MultiSimDiff (ours)** + U-Net | 0.0119 | **0.0141** | 0.0046 | **0.0174** |

Table 1 presents the relative L2 norm (L2 norm of prediction error divided by L2 norm of the ground-truth) in predictions made by surrogate model and MultiSimDiff on a validation set of decoupled data and a test set of coupled data. For FNO, the prediction error for $u$ is comparable between MultiSimDiff and surrogate models on decoupled data; however, MultiSimDiff shows a significantly larger error in predicting $v$, which is four times that of the surrogate model. As a result, the error in predicting the coupled solution for $v$ is greater than that of the surrogate model, while the error for $u$ is lower. For U-Net, the prediction errors for $u$ and $v$ are similar between MultiSimDiff and surrogate models on decoupled data, but MultiSimDiff achieves a lower error for the coupled solution.

This straightforward experiment tests the correctness of MultiSimDiff but shows no significant advantages over the surrogate model. However, the surrogate model fails in solving more complex problems, which will be discussed in the next section.

## 4.2 NUCLEAR THERMAL COUPLING

This experiment tests the performance of MultiSimDiff in more physical processes and coupling modes, including both regional and interface coupling, strong coupling and weak coupling, unidirectional and bidirectional coupling. We focus on nuclear thermal coupling in transient conditions

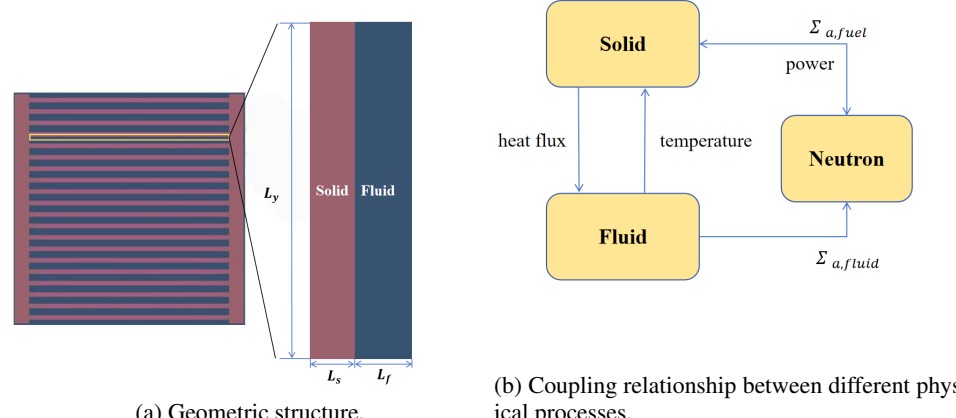

(a) Geometric structure.

(b) Coupling relationship between different physical processes.

Figure 2: Problem description of nuclear thermal coupling.

for plate fuel elements. To simplify, a typical pin cell is analyzed (as shown in Fig. 2a), and the transient disturbance is modeled as a change in neutron flux density at the boundary. We aim to solve the neutron physics equation across the entire domain, the heat conduction equation in solid, and the flow heat transfer equations in fluid. This problem involves conjugate heat transfer between solid and fluid phases, presenting an interface coupling issue. Additionally, the negative feedback between the neutron physics field and the temperatures of the fluid and solid introduces a regional coupling. Aside from the unidirectional coupling from the fluid fields to the neutron physics field, all other interactions are bidirectional. The neutron physics field is weakly coupled with the temperature fields of solid and fluid, while the coupling effect at the interface between solid and fluid is strong. The objective is to predict the evolution of the entire physical system under different neutron boundary conditions. The coupling relationship between different physical processes are shown in Fig. 2b.

Generating estimated physical fields in this two-dimensional time series problem with three physical processes is challenging using Gaussian random fields. To address this, we employ a pre-iteration method for data generation. The validation dataset consists of decoupled data not used during training, while the test dataset comprises coupled data. Coupled data is computed using the operator splitting iterative algorithm (MacNamara & Strang, 2016), which exchanges information between physical processes at each time step. Additional details on the datasets, governing equations, network architecture, and training process can be found in Appendix C.

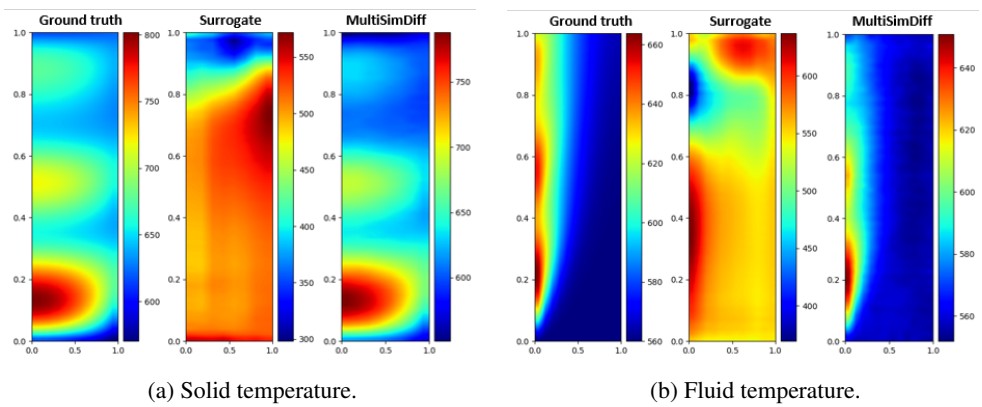

(a) Solid temperature.      (b) Fluid temperature.

Figure 3: Comparison of prediction results between MultiSimDiff and surrogate model. The surrogate model fails on the test set of the coupled scenario.

Table 2 displays the relative prediction errors of surrogate models and MultiSimDiff on a validation set of decoupled data and a test set of coupled data. In single physical process prediction (decoupled data), surrogate models outperform MultiSimDiff. However, in predicting the coupled solution, all surrogate models fail except for the neutron physics field, with the predicted solid and fluid temper-

Table 2: Relative L2 norm of prediction error on nuclear thermal coupling for multiphysics simulation. The unit is $1 \times 10^{-2}$.

| method | neutron | | solid | | fluid | |
|---|---|---|---|---|---|---|
| | decoupled | coupled | decoupled | coupled | decoupled | coupled |
| surrogate + FNO | 0.251 | 22.1 | 0.0445 | 31.8 | 0.106 | 10.2 |
| **MultiSimDiff (ours)** + FNO | 0.738 | **8.42** | 0.175 | **9.72** | 0.615 | **7.31** |
| surrogate + U-Net | 0.181 | 4.45 | 0.0800 | 18.2 | 0.0927 | 8.03 |
| **MultiSimDiff (ours)** + U-Net | 0.487 | **1.97** | 0.108 | **2.87** | 0.303 | **3.91** |

ature fields shown in Fig. 3 (for more visualizations, see Fig. 5). The neutron physics field remains relatively accurate because the feedback from solid and fluid temperatures is weak and primarily driven by external input boundary conditions. In contrast, solid temperature and fluid fields are significantly influenced by other physical processes , leading to non-physical predictions due to the lack of iterative process data during training. In comparison, MultiSimDiff more accurately captures the morphology of coupled solutions and demonstrates higher accuracy. In addition, we further use DDIM (Song et al., 2021) to accelerate sampling and compare the operational efficiency of different methods. MultiSimDiff achieves an acceleration of up to 29 times, with detailed information in Appendices H and I.

### 4.3 PRISMATIC FUEL ELEMENT

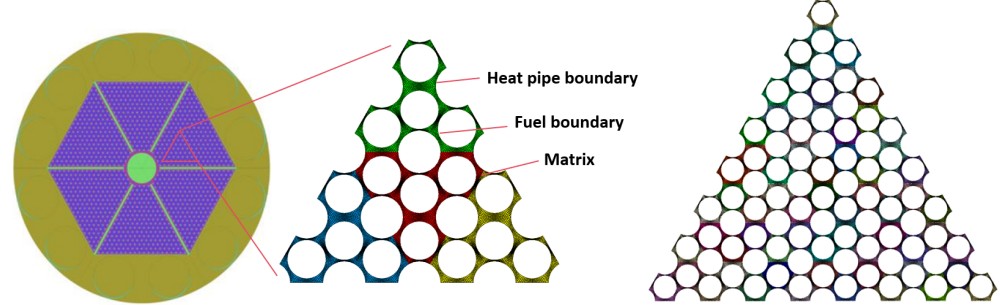

(a) The structure used to generate training data.     (b) Larger structure for testing.

Figure 4: Problem setup of the prismatic fuel element. In (a), the left figure shows the entire reactor, with the purple section representing the reactor core, which mainly contains fuel elements, a matrix, and heat pipes. The right figure illustrates a portion of the reactor core, displaying only the matrix while omitting the heat pipes and fuel elements. This structure composed of 16 fuel elements is used to generate training data. (b) is a large structure composed of 64 elements used for testing.

This experiment tests the ability of MultiSimDiff to solve multi-component simulation problems, focusing on the thermal and mechanical analysis of prismatic fuel elements for a new type of reactor (Ma et al., 2022), as shown in Fig. 4. The reactor core consists of three components: fuel, matrix, and heat pipe. Since engineering focuses mainly on the matrix, we consider the fuel and fluid as boundary conditions for analysis. Different heat fluxes will be assigned to the fuel boundary to simulate various heat release behaviors of the fuel rods. The aim is to train a model that predicts its temperature $T$ and strain $\varepsilon_x, \varepsilon_y$ based on the solutions of its three neighbors and its heat flux, and then use this basic model to predict larger structure as shown in Fig. 4b.

The training data originate from a medium structure simulation that includes 16 fuel elements, as shown in Fig. 4a; hence, a single simulation data point can generate 16 training data. The well-trained model will be tested on two structures: one is the medium structure used for data generation, and the other is a large structure containing 64 fuel elements. Further details on the datasets, network architecture, and training process are provided in Appendix D.

Table 3 presents the prediction relative errors of surrogate model and MultiSimDiff across three tasks: a single fuel element, a medium structure of 16 fuel elements, and a large structure of 64 fuel elements. The average relative error of strain $\varepsilon_x, \varepsilon_y$ is denoted as $\varepsilon$. GIN and SAN learn on small

Table 3: Relative L2 norm of prediction error on prismatic fuel element experiment, for single-component and multi-component simulation. The multi-component includes 16-component (medium) and 64-component (large) simulations. The unit is $1 \times 10^{-2}$.

| method | single | | 16-component | | 64-component | |
|---|---|---|---|---|---|---|
| | $T$ | $\varepsilon$ | $T$ | $\varepsilon$ | $T$ | $\varepsilon$ |
| GIN | - | - | 1.96 | 3.18 | 4.63 | 7.02 |
| SAN | - | - | 0.114 | 16.5 | $1.00\times10^2$ | $1.18\times10^4$ |
| surrogate + Geo-FNO | 0.0883 | 0.195 | **0.337** | 2.59 | divergent | divergent |
| **MultiSimDiff (ours)** + Geo-FNO | 0.139 | 0.459 | 0.338 | **2.42** | **0.950** | **3.52** |
| surrogate + Transolver | 0.0764 | 0.251 | 0.314 | 1.13 | 1.25 | 3.31 |
| **MultiSimDiff (ours)** + Transolver | 0.107 | 0.303 | **0.213** | **1.03** | **0.759** | **1.94** |

graphs with 16 components and test on large graphs with 64 components. Due to the uniformity of graph structures in all training data and the fact that SAN learns a global relationship, SAN fails to predict larger structures. In contrast, GIN, capable of learning a local relationship, succeeds in handling larger structures. However, when compared to the surrogate model and MultiSimDiff, GIN has a larger error.

Subsequently, a comparative analysis between the surrogate model and MultiSimDiff has been conducted. The surrogate model performs better in predicting a single component, but for medium structure, MultiSimDiff outperforms it. It's important to note that the surrogate model's predictions occasionally diverge, necessitating adjustments to the relaxation factor to maintain stability. For the large structure, U-Net in the surrogate model demonstrates better stability, while the FNO model continues to diverge even after relaxation factor adjustments. MultiSimDiff is very stable and accurate, and no divergence phenomenon has been observed. Compared with the surrogate model, the relative error of MultiSimDiff has been reduced by 40.3% on average, demonstrating its ability to generalize to much larger multi-component simulations while trained on single components. In addition, we further use DDIM to accelerate sampling and compare the operational efficiency of different methods.MultiSimDiff achieves an acceleration of up to 41 times, with detailed information in Appendices H and I.

## 5   LIMITATION AND FUTURE WORK

There are also several limitations of our proposed MultiSimDiff that provide exciting opportunities for future work. Firstly, in multiphysics simulation, although the MultiSimDiff trained on decoupled data can predict coupled solutions more accurately than baseline surrogate models, the prediction errors are still higher compared to single physical processes predictions. In addition, there is a certain gap in accuracy compared to models trained through coupled data, as shown in Appendix E. Future efforts can focus on improving dataset generation, training methods, and incorporating physical information to boost accuracy. Secondly, we plan to explore additional accelerated sampling algorithms, aiming to significantly improve efficiency while maintaining prediction accuracy. Lastly, the experiments in this paper simplify many aspects compared to real engineering problems, and future work will aim to validate the algorithm in more complex real-world scenarios.

## 6   CONCLUSION

This work presents MultiSimDiff as a novel method for multiphysics and multi-component simulations. In multiphysics scenarios, models trained on decoupled data can predict coupled solutions, while in multi-component simulations, models trained on small structures can extrapolate to larger ones. We develop three datasets to validate MultiSimDiff and compare it to the surrogate model method. Results show that MultiSimDiff effectively predicts coupled solutions in multiphysics simulations where surrogate models fail, and exhibits greater accuracy in predicting larger structures in multi-component simulations. We believe this approach provides a new approach to address multiphysics and multi-component simulations, important across science and engineering.

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

## A    ALGORITHM FOR MULTI-COMPONENT SIMULATION.

Multi-component simulation first requires training a diffusion model to predict the solution of the current component based on the solutions of its neighboring components. Additionally, it is necessary to define the connectivity of all components and the function $f$ to update the surrounding components' solutions for each component. The multi-component simulation algorithm is presented in Algorithm 2. Lines 6 to 11 are the denoising cycle of the diffusion model, in each diffusion step, the solutions of each component are updated together.

---

**Algorithm 2** Algorithm for multi-component simulation by MultiSimDiff

---

**Require:** A diffusion model $\epsilon_\theta(z_{\partial v_i}, C, s)$, outer inputs $C$, diffusion step $S$, number of external loops $K$, number of component $N$, connectivity of all components adj, update function $f(z_{v_1}, ..., z_{v_n}, \text{adj})$ of $z_{\partial v}$.

1: $z_{v_i}^e \sim \mathcal{N}(0, I)$ // initialize estimated solution for each component $v_i$
   // Add an external loop to improve $z_{v_i}^e$:
2: **for** $k = 1, ..., K$ **do**
3:     $\hat{z}_{v_i}^e \leftarrow z_{v_i}^e$ // update previous estimated solutions for each component $\hat{z}_{v_i}^e$
4:     $z_{v_i}^e \sim \mathcal{N}(0, \mathbf{I})$ // initialize current estimated solutions for each component $z_i^e$
5:     $z_{v_i, S} \sim \mathcal{N}(0, \mathbf{I})$ // initialize solutions for each component $z_{v_i}$
       // denoising cycle of diffusion model:
6:     **for** $s = S, ..., 1$ **do**
7:       $\lambda = 1 - \frac{s}{S}$ if $k > 1$ else 1 // define the weights of $\hat{z}_i^e$ and $z_i^e$
8:       $z_{\partial v} = f(z_{v_1}^e, ..., z_{v_n}^e, \text{adj})$ // update the solutions of surrounding components for each $z_{v_i}$
9:       $w \sim \mathcal{N}(0, \mathbf{I})$
         // use weighted estimated solutions as conditions for single step denoising,
         // update all components together:
10:      $z_{v_i, s-1} = \frac{1}{\sqrt{\alpha_s}}(z_{v_i, s} - \frac{1-\alpha_s}{\sqrt{1-\overline{\alpha}_s}}\epsilon_\theta(z_{v_i, s} \mid \lambda z_{\partial v_i}^e + (1-\lambda)\hat{z}_{\partial v_i}^e, C)) + \sigma_s w$
         // update the estimated solutions of all components together
11:      $z_{v_i}^e = \frac{1}{\sqrt{\overline{\alpha}_s}}(z_{v_i, s} - \sqrt{1-\overline{\alpha}_s}\epsilon_\theta(z_{v_i, s} \mid \lambda z_{v_i}^e + (1-\lambda)\hat{z}_{\partial v_i}^e, C))$
12:     **end for**
13: **end for**
14: **return** $z_{v_i, 0}$

---

## B   Additional details for reaction-diffusion

This section provides additional details for Section 4.1.

**Problem description.** The 1D FitzHugh Nagumo reaction diffusion equation takes the form:

$$
\begin{cases}
\dfrac{\partial u}{\partial t} = \mu_u \Delta u + u - u^3 - v + \alpha, x \in [0,1], t \in [0,5] \\
\dfrac{\partial v}{\partial t} = \mu_v \Delta v + (u-v)\beta, x \in [0,1], t \in [0,5] \\
[u,v] = [u_0, v_0], x \in [0,1], t = 0
\end{cases}
\tag{15}
$$

The coefficients $\mu_u, \mu_v, \alpha$, and $\beta$ are set to 0.01, 0.05, 0.1, and 0.25, respectively.

**Dataset.** We employed the solve_ivp function in Python to solve the reaction-diffusion equations. The spatial mesh consisted of $n_x = 20$ points, the time step is adaptively controlled by the algorithm, but only outputs the results of 10 time steps. To train the data for a single physical process, it was necessary to assume the initial conditions of the other physical processes and the current field. For instance, training $u$ required assumptions about $u_0$ and $v$. The dimension of $u_0$ is $[n_x]$, which was generated using a one-dimensional Gaussian random field, and $v$ has dimensions $[n_t, n_x]$, and was generated by sampling a one-dimensional Gaussian random field $n_t$ times.

**Model structure.** The 2D U-Net and 2D FNO serve as both the surrogate and MultiSimDiff. U-Net consists of modules: a downsampling encoder, a middle module, and an upsampling decoder. The encoder and decoder comprise four layers, each with three residual modules and downsampling/upsampling convolutions, with the third module incorporating attention mechanisms. The middle module also contains three residual modules, with attention mechanisms included in the second module. The input data is encoded into a hidden dimension before undergoing sequential downsampling and upsampling. FNO consists of three modules: a lift-up encoder, $n$ FNO layers, and a projector decoder. Each FNO layer includes a spectral convolution, a spatial convolution, and a layer normalization. The surrogate model predicts the evolution of the current physical process using its initial conditions and those of other physical processes. Its input dimension is [b, 1, 10, 20] and output dimension is [b, 1, 10, 20]. The diffusion model has an input dimension of [b, 2, 10, 20] and an output dimension of [b, 1, 10, 20], with b representing the batch size. The shape of

---

**Algorithm 3** Surrogate model combination algorithm.

---

**Require:** Compositional set of surrogate model $\epsilon_\theta^i(z_{\neq i}, C), i = 1, 2, ..., N$, outer inputs $C$, maximum number of iterations $M$, tolerance $\epsilon_{max}$, relaxation factor $\alpha$.
    Initialize constant fields $z_i$, $m = 0$
    **while** $m < M$ and $\epsilon > \epsilon_{max}$ **do**
        $m = m + 1$
        $\hat{z}_i = z_i$
        **for** $i = 1, ..., N$ **do**
            $z_i = \alpha\epsilon_\theta^i(z_{\neq i}, C) + (1 - \alpha)\hat{z}_i$
        **end for**
        $\epsilon = L_1(z_i - \hat{z}_i)$
    **end while**
    **return** $z_i$

---

initial condition of [b, 1, 1, 20] and will repeat to align the required shape. The diffusion step of the diffusion model is set to 250. More details are shown in Table 4.

**Training.** The surrogate model and MultiSimDiff are trained similarly, with further details in Table 5.

**Inference.** The hyperparameter $K$ is set to 2. The surrogate models' combination algorithm in experiments 1 and 2 is identical, as demonstrated in Algorithm 3. The relaxation factor $\alpha$ is set to 0.5.

Table 4: Hyperparameters of model architecture for reaction-diffusion task.

| Hyperparameter name | $u$ | $v$ |
|---|---|---|
| Hyperparameters for U-Net architecture: | | |
| Channel expansion factor | (1,2) | (1,2) |
| Number of downsampling layers | 2 | 2 |
| Number of upsampling layers | 2 | 2 |
| Number of residual blocks for each layer | 3 | 3 |
| Hidden dimension | 24 | 24 |
| Hyperparameters for FNO architecture: | | |
| FNO width | 24 | 24 |
| number of FNO layer | 4 | 4 |
| FNO mode | [6,12] | [6,12] |
| padding | [8,8] | [8,8] |

Table 5: Hyperparameters of training for reaction-diffusion task.

| Hyperparameters for U-Net and FNO training | $u \& v$ |
|---|---|
| Loss function | MSE |
| Number of examples for training dataset | $10^4$ |
| Total number of training steps (surrogate; diffusion) | $10^5; 2 \times 10^5$ |
| Gradient accumulate every per epoch | 2 |
| learning rate | $10^{-4}$ |
| Batch size | 256 |

## C ADDITIONAL DETAILS FOR NUCLEAR THERMAL COUPLING

This section provides additional details for Section 4.2.

**Problem description.** The goal of this problem is to predict the performance of plate-type fuel assembly under transient conditions. A typical pin cell in JRR-3M fuel assembly (Gong et al., 2015) is adopted as the computational domain, as shown in Fig. 2. For simplicity, the cladding in the

fuel plate is omitted here without losing the representativeness of its multiphysics coupling feature. U-Zr alloy and lead-bismuth fluid are adopted as fuel and coolant materials, respectively. Their physical property parameters can be found in the anonymous repository. We consider a single-group diffusion equation for the neutron physics process and employ an incompressible fluid model for coolant modeling. Temperature fields in solid and fluid can influence the macroscopic absorption cross-section in the neutron physics equation, while neutron flux affects the heat source in the fuel domain. Conjugate heat transfer occurs at the interface between the fluid and solid domains. While the feedback of temperature on neutrons is inherently complex, a linear negative feedback is assumed for simplicity. The governing equations are presented in Eq. 16, Eq. 17, and Eq. 18.

$$
\begin{cases}
\dfrac{1}{v}\dfrac{\partial \phi(x,y,t)}{\partial t} = D\Delta\phi + (v\Sigma_f - \Sigma_a(T))\phi, x \in [0, L_s + L_f], y \in [0, L_y], t \in [0,5] \\
\phi(0,y,t) = f(y,t) \\
\phi(L_s + L_f, y, t) = \phi(x,0,t) = \phi(x, L_y, t) = 0
\end{cases}
\tag{16}
$$

$$
\begin{cases}
\dfrac{\rho c_p T_s(x,y,t)}{\partial t} = \nabla k_s \nabla T_s + A\phi_s, x \in [0, L_s], y \in [0, L_y], t \in [0,5] \\
\dfrac{\partial T_s(x,0,t)}{\partial y} = \dfrac{\partial T_s(x, L_y, t)}{\partial y} = 0 \\
T_s(L_s, y, t) = T_f(L_s, y, t)
\end{cases}
\tag{17}
$$

$$
\begin{cases}
\nabla \cdot \vec{u} = 0, x \in [L_s, L_s + L_f], y \in [0, L_y], t \in [0,5] \\
\rho\left(\dfrac{\partial \vec{u}}{\partial t} + \vec{u} \cdot \nabla \vec{u}\right) = -\nabla p + \mu \nabla^2 \vec{u} + \vec{f}, x \in [L_s, L_s + L_f], y \in [0, L_y], t \in [0,5] \\
\rho c_p\left(\dfrac{\partial T_f}{\partial t} + \vec{u} \cdot \nabla T_f\right) = k_f \nabla^2 T, x \in [L_s, L_s + L_f], y \in [0, L_y], t \in [0,5] \\
k_f \dfrac{\partial T_f(L_s, y, t)}{\partial x} = k_s \dfrac{\partial T_s(L_s, y, t)}{\partial x}
\end{cases}
\tag{18}
$$

Here $v$ is neutrons / per fission, $D$ is the diffusion coefficient of the neutron, $\Sigma_f, \Sigma_a$ are the fission and absorption cross-section, respectively, and we only consider the feedback of temperature on the absorption cross-section $\Sigma_a$ here. $k_s, k_f$ are the conductivity of solid and fluid, respectively, both being functions of $T$.

**Dataset.** We utilize the open-source finite element software MOOSE (Multiphysics Object-Oriented Simulation Environment) (Icenhour et al., 2018) to tackle the nuclear thermal coupling problem. The solid temperature field uses a mesh of [64,8], the fluid fields have a mesh of [64,12], and the neutron physics field employs a mesh of [64,20]. The neutron physics and solid temperature fields are calculated using the finite element method at mesh points, while the fluid domain uses the finite volume method at mesh centers. Interpolation is applied to align the neutron physics and solid temperature values with the fluid fields. The time step is adaptively controlled by the algorithm, but only outputs the results of 16 time steps. So the input dimensions for the surrogate models of neutron physics field, solid temperature field, and fluid fields are [b,2,16,64,20], [b,2,16,64,8], and [b,1,16,64,12], respectively. The input dimensions for the diffusion model of the three fields are [b,3,16,64,20], [b,3,16,64,8], and [b,5,16,64,12], respectively. The output dimensions of the three fields are [b,1,16,64,20], [b,1,16,64,8], and [b,4,16,64,12], respectively.

As noted, assuming the distribution of physical field data in high-dimensional problems is challenging. We recommend a pre-iteration method for data generation. Initially, we assume constant values for all other physical fields and calculate the current field. This process repeats until all fields are computed. If there are $n$ physical fields, pre-iteration requires $n$ - 1 calculations plus one iteration for data generation, totaling $2n$ - 1 calculations. To accelerate data generation, the most time-consuming field can be excluded from pre-iteration. In this problem, the fluid fields' computation time is approximately three times that of the other fields, so it is excluded from pre-iteration. The process begins by assuming constant fluid fields and solid temperatures to calculate the neutron physics field, followed by using the resulting neutron physics field and assumed fluid fields' temperature to calculate the solid temperature. The data generation proceeds sequentially with calculations for the fluid fields, neutron physics field, and solid temperature field.

**Model structure.** The 3D U-Net and 3D FNO serve as both the surrogate model and MultiSimDiff, using a layer design identical to the 2D. For regional coupling, concatenation is directly applied to the channel dimension using the concat function. In contrast, for interface coupling, dimensions must be replicated to align spatially before concatenation. The conditioning of the diffusion step for FNO is operating in spectrum space (Gupta & Brandstetter, 2023), which is better than in the original space for this problem. The diffusion step of the diffusion model is set to 250. More details are shown in Table 6.

**Training.** The surrogate model and MultiSimDiff are trained similarly, but training neutron physics fields using MultiSimDiff requires more time to converge, with further details in Table 7.

**Inference.** The hyperparameter $K$ is set to 2. The relaxation factor for surrogate model $\alpha$ is set to 0.5.

**Detailed results.** Fig. 5 presents the results of predicting various physical fields using the last time step surrogate model and MultiSimDiff + U-Net on the final test data. The neutron physics field and solid temperature field are represented by $\phi$ and $T_s$, respectively. The fluid fields include four physical quantities: $T_f, P, u_x, u_y$, totaling six quantities. Since the neutron physics field and the $u_y$ component of the fluid fields are less influenced by other physical processes, the surrogate model can still make predictions, but the accuracy is lower than that of the MultiSimDiff. Besides, the surrogate model has failed to predict the other physical processes. In contrast, MultiSimDiff continues to provide relatively accurate predictions, although some distortions are observed in certain regions.

Table 6: Hyperparameters of model architecture for nuclear thermal coupling task.

| Hyperparameter name | neutron | solid | fluid |
|---|---|---|---|
| Hyperparameters for U-Net architecture | | | |
| Channel Expansion Factor | (1,2,4) | (1,2,4) | (1,2,4) |
| Number of downsampling layers | 3 | 3 | 3 |
| Number of upsampling layers | 3 | 3 | 3 |
| Number of residual blocks for each layer | 3 | 3 | 3 |
| Hidden dimension | 8 | 8 | 16 |
| Hyperparameters for FNO architecture | | | |
| FNO width | 8 | 8 | 16 |
| number of FNO layer | 3 | 3 | 3 |
| FNO mode | [6,16,8] | [6,16,4] | [6,16,6] |
| padding | [8,8,8] | [8,8,8] | [8,8,8] |

Table 7: Hyperparameters of training for nuclear thermal coupling task.

| Hyperparameters for U-Net and FNO training | neutron&solid&fluid |
|---|---|
| Loss function | MSE |
| Number of examples for training dataset | $5 \times 10^3$ |
| Total number of training steps (surrogate; diffusion) | $10^5; 2 \times 10^5$ |
| Gradient accumulate every per epoch | 2 |
| learning rate | $10^{-4}$ |
| Batch size | 32 |

## D  ADDITIONAL DETAILS FOR PRISMATIC FUEL ELEMENT

This section provides additional details for Section 4.3.

**Problem description.** This problem aims to predict the thermal and mechanical performance of prismatic fuel elements in heat pipe reactor Ma et al. (2022) at different source power. The reactor core is stacked up using a hexagonal prism SiC matrix, with multiple holes dispersed in the matrix for containing fuel elements and heat pipes as shown in Fig. 6. The SiC matrix plays a role in locating the fuel and heat pipes at expected positions in the core. The entire structure consists of two

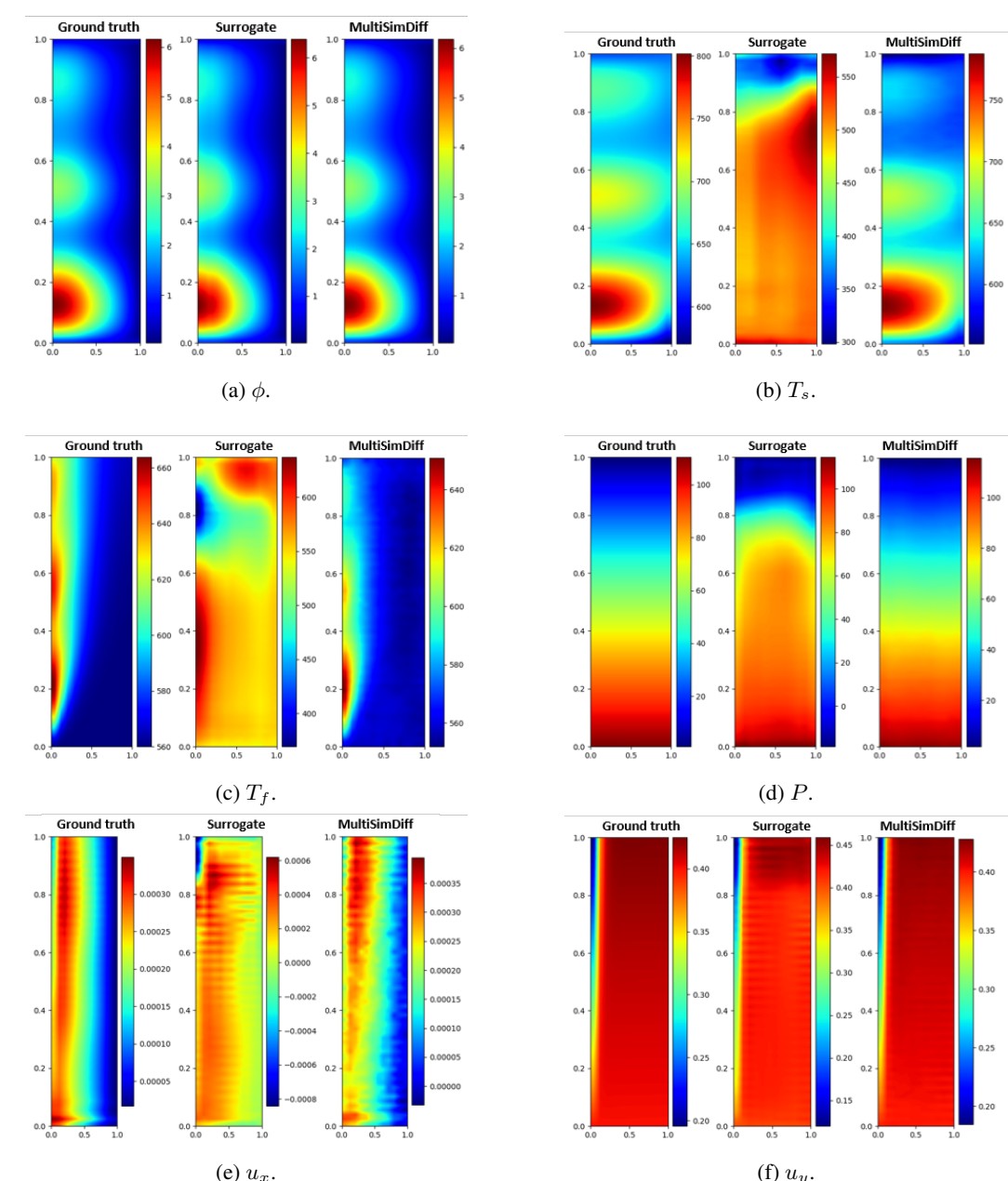

Figure 5: Comparison of surrogate model and MultiSimDiff for predicting all physical fields.

basic components, one oriented upwards and the other downwards, as illustrated in Fig. 7. Fission energy released in fuel elements is dissipated using heat pipes. Both the fuel elements and heat pipes are considered as boundaries here, and only the more concerned matrix behavior is analyzed in the demonstration. Only strain is predicted here since stress can be derived from the mechanical constitutive equation, and displacement is obtained through strain integration. The analysis uses the plane strain assumption ($\varepsilon_z = 0$) and excludes irradiation effects, simplifying it to a steady-state problem.

**Dataset.** We use MOOSE to calculate the thermal and mechanical problems. The training data comes from a medium structure simulation with 16 fuel elements, allowing each simulation to generate 16 training data, as shown in Fig. 4. This structure is chosen because a fundamental component, along with its neighboring components, is entirely contained within the interior, which is

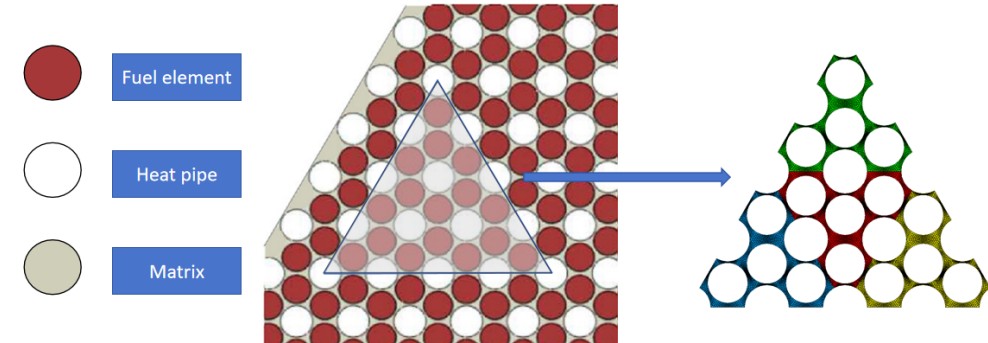

Figure 6: Schematic of heat pipe reactor core structure. The left figure shows a partial structure of the entire reactor, with multiple holes dispersed in the matrix for containing fuel elements and heat pipes. The right figure shows how to select a medium structure for analysis from the overall structure.

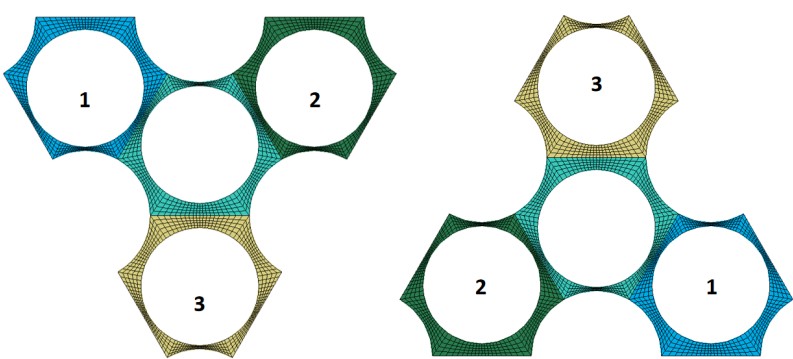

Figure 7: Two basic components: one facing upwards (left) and the other facing downwards (right).

where most components that need to be predicted in large structures are located. When generating data, the heat flux density is uniformly sampled from the range $[10^5, 10^6]$ W/m. A free boundary condition is randomly assigned to one edge, while symmetric boundary conditions are applied to the remaining two edges. Each fundamental component is uniformly meshed with 804 points, each requiring the prediction of three physical quantities. To predict the central component, the heat flux density of this component and the coordinates of each mesh point are concatenated with data from its three neighboring components, yielding an input dimension of [b, 804, 15] for the diffusion model and [b, 804, 12] for the surrogate model, where b is the batch size. The output dimension is [b, 804, 3]. The sequence of neighboring elements is consistent, with the downward-facing center element being the upward-facing center element rotated by 180 degrees. This arrangement is illustrated in Fig. 7. Boundary conditions are considered only for symmetric and free types, represented as [0, 1, 1] and [0, 0, 0], respectively, and are replicated to a dimension of [804, 3].

**Model structure.** The Geo-FNO and Transolver serve as both the surrogate model and MultiSimD-iff. Geo-FNO enhances FNO for irregular meshes using three modules: a geometry encoder that converts physical fields from irregular to latent uniform meshes, FNO functioning in latent space, and a geometry decoder that transforms physical fields from the uniform mesh back to the original irregular mesh. We utilize a 2D Geo-FNO that transforms into a 2D uniform mesh. Transolver is designed to tackle complex structural simulation problems involving numerous mesh points by learning the intrinsic physical states of the discretized domain. Given a mesh set with $N$ points and $C$ features per point, the network first assigns each mesh point to $M$ potential slices, transforming the shape from $N \times C$ to $M \times N \times C$. It then applies spatially weighted aggregation, resulting in a shape of $M \times C$. Self-attention is used to capture intricate correlations among different slices, after which the data is transformed back to the mesh points. The conditioning of diffusion step for

Geo-FNO is also operating in spectrum space (Gupta & Brandstetter, 2023). More details about the network can be found in (Wu et al., 2024a). The setting of hyperparameters is shown in Table 8. The diffusion step of the diffusion model is set to 250.

**Training.** The surrogate model and MultiSimDiff are trained similarly, but training MultiSimDiff requires more time to converge, with further details in Table 9.

**Inference.** This problem uses the same neural network to predict the performance of all elements, allowing for simultaneous updates of the physical fields and enhancing inference speed. This method applies to both diffusion and surrogate models. The hyperparameter $K$ is set to 3.

**Detailed results.** Fig. 8 and Fig. 9 compares the results of predicting the large structure using the surrogate model and MultiSimDiff + U-Net. Because the surrogate model of FNO fails in predicting large structures, only the results of MultiSimDiff + FNO are provided in Fig. 10. The strain is only displayed in the $x$-direction due to its similarity in both $x$ and $y$. The error graph indicates that MultiSimDiff offers more accurate predictions.

**Graph neural network and Graph Transformer configuration.** For GIN and SAN, each component is treated as a node in the graph, with training conducted on a small graph of 16-component, and ultimately tested on a larger graph of 64-component. Compared with the surrogate model and MultiSimDiff, they use only the system's input as input features. In contrast, the surrogate model and MultiSimDiff enrich its input by incorporating the solutions from the surrounding component, thereby improving accuracy, as demonstrated in Table 3. The input to the GIN and SAN is the heat flux density and boundary conditions of each component, and the output is the physical quantities at all grid points on the component. GIN updates the nodes on the graph through the graph structure, whereas SAN captures graph structural information by inputting the eigenvalues and eigenvectors of the graph Laplacian matrix into a transformer. The training settings of GIN and SAN are consistent with the MultiSimDiff. We have adjusted the number of network layers and the size of hidden layers to obtain the model with optimal performance.

Table 8: Hyperparameters of model architecture for prismatic fuel element task.

| Hyperparameter name | |
|---|---|
| Hyperparameters for Transolver | |
| Number of layers | 5 |
| Number of head | 8 |
| Number of slice | 16 |
| Hidden dim | 64 |
| Hyperparameters for Geo-FNO | |
| Uniform grid size | [64, 64] |
| FNO width | 5 |
| FNO mode | [8,8] |
| Number of FNO layer | 3 |
| Hidden dim | 64 |

Table 9: Hyperparameters of training for prismatic fuel element task.

| Hyperparameters for Transolver and Geo-FNO training | |
|---|---|
| Loss function | MSE |
| Number of examples for training dataset | 16000 |
| Total number of training steps(surrogate;diffusion) | $10^5; 2 \times 10^5$ |
| Gradient accumulate every per epoch | 2 |
| learning rate | $10^{-4}$ |
| Batch size | 256 |

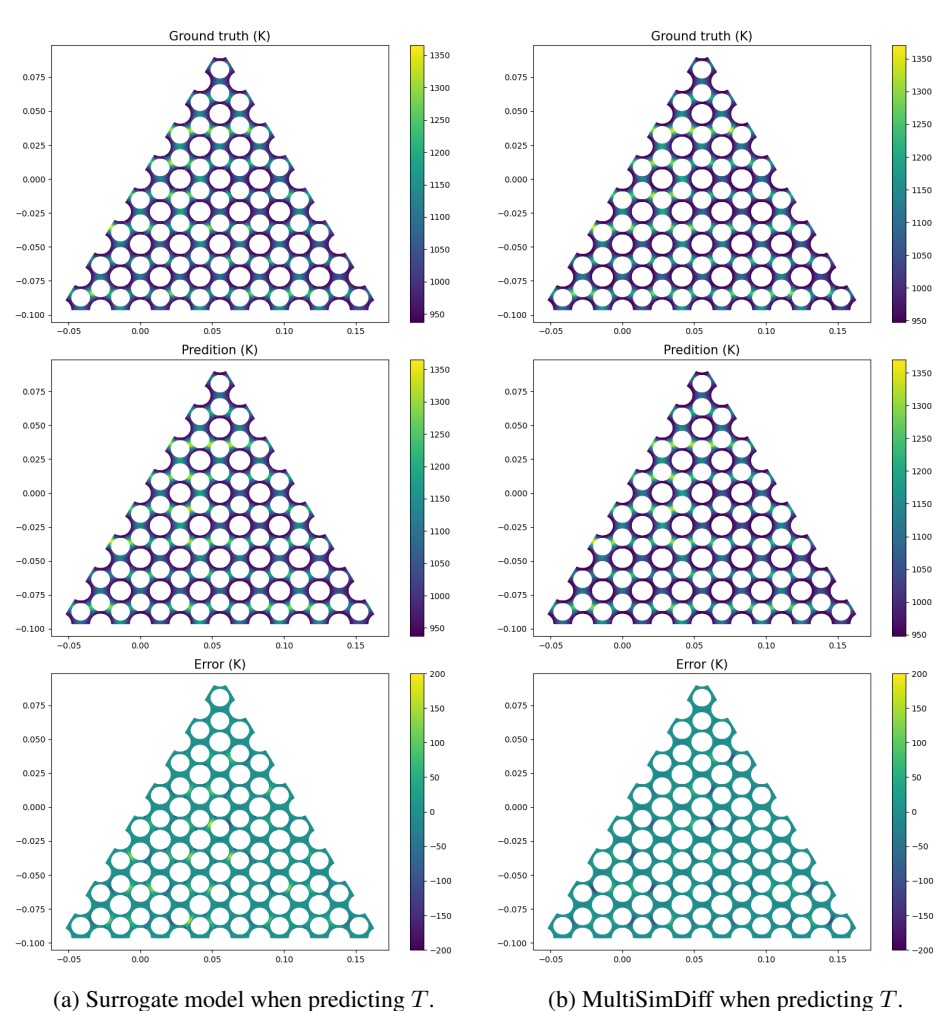

(a) Surrogate model when predicting $T$.      (b) MultiSimDiff when predicting $T$.

Figure 8: Comparison of surrogate models and MultiSimDiff + U-Net for predicting the temperature of large structures.

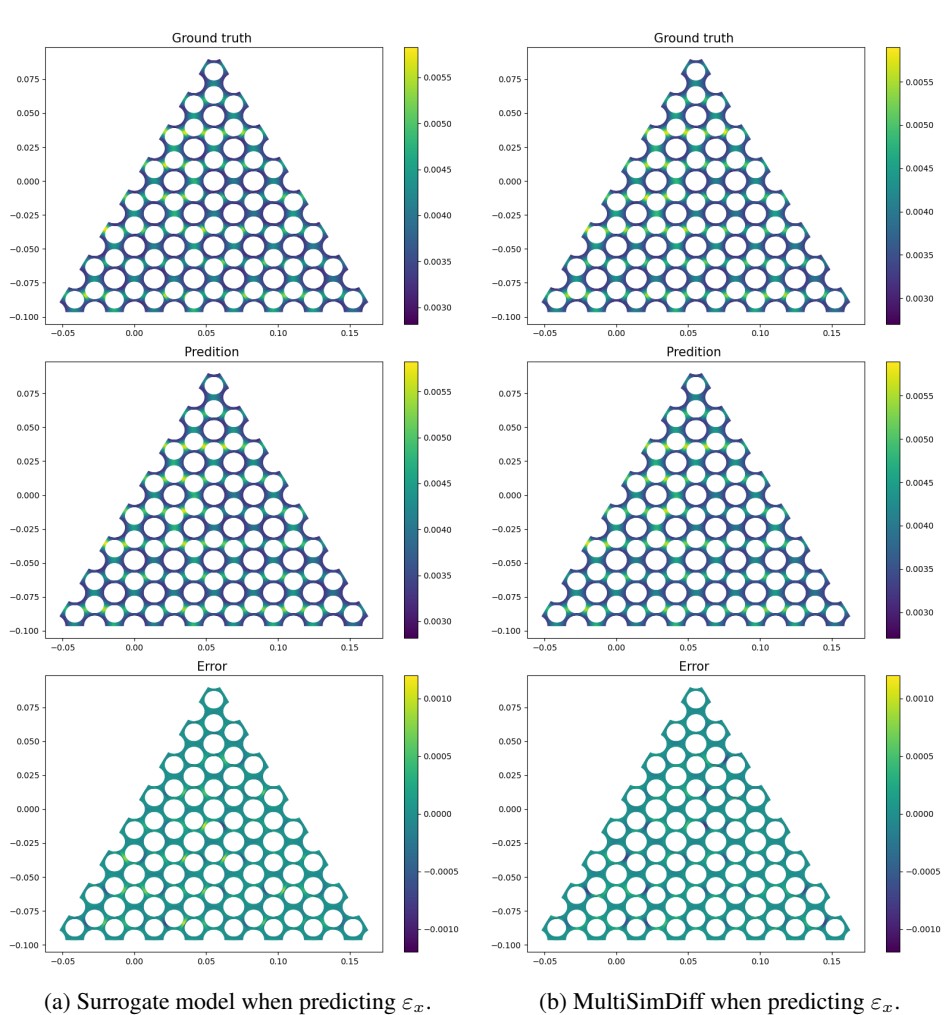

(a) Surrogate model when predicting $\varepsilon_x$.   (b) MultiSimDiff when predicting $\varepsilon_x$.

Figure 9: Comparison of surrogate models and MultiSimDiff + U-Net for predicting the strain of large structures.

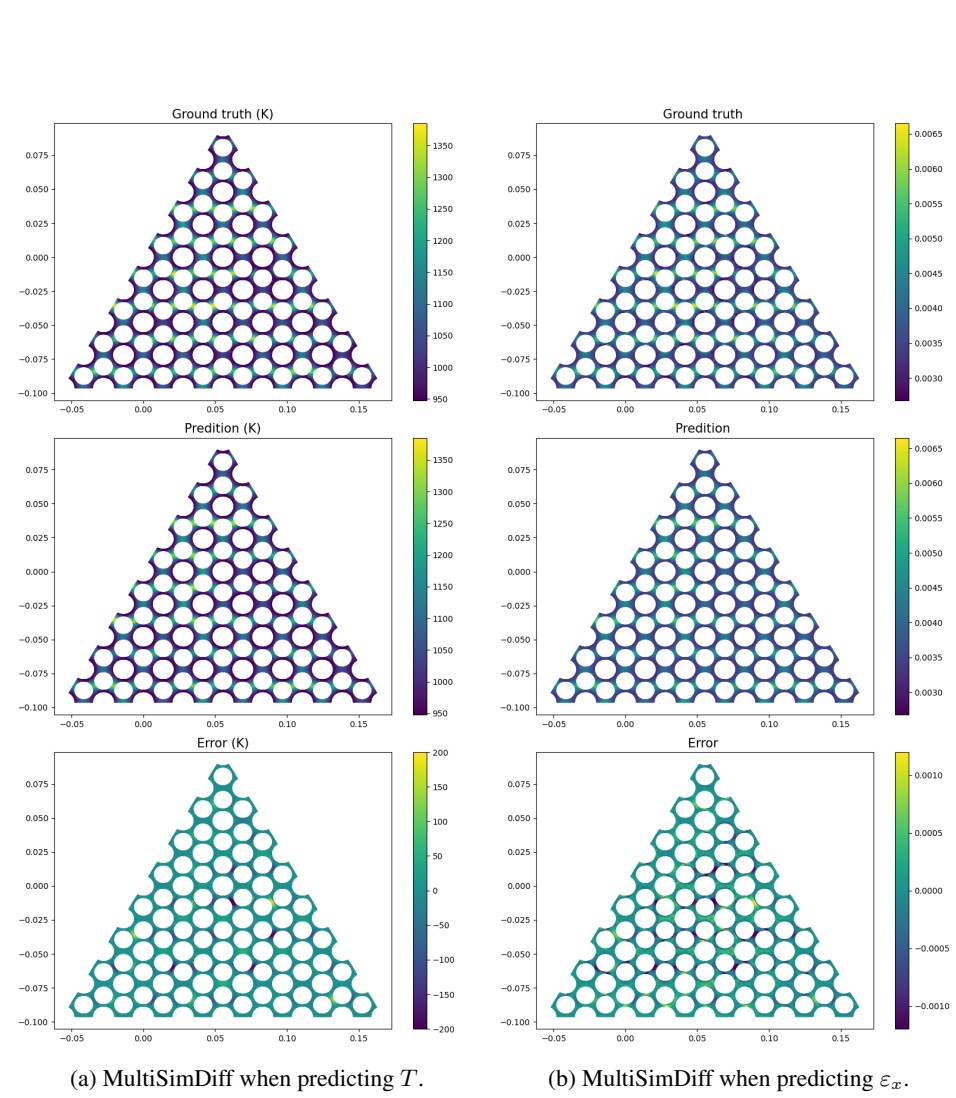

(a) MultiSimDiff when predicting $T$.

(b) MultiSimDiff when predicting $\varepsilon_x$.

Figure 10: The results of MultiSimDiff + FNO for predicting large structure.

# E COMPARISON OF MODELS TRAINED USING COUPLED AND DECOUPLED DATA.

To further investigate the model's boundaries for multiphysics simulation, we utilize coupled data to train diffusion models and compare them to models trained on decoupled data in experiments 1 and 2. The input of the diffusion model is the external input of the physical system, while the output is the solution of the coupled physical fields. In experiment 1, the input consists of the initial conditions of $u$ and $v$, with the output being their trajectories. Since $u$ and $v$ are defined on the same grid, a single network can be employed to predict $u$ and $v$ together. In experiment 2, the input is the variation of neutron boundaries over time, and the output is the trajectories of the neutron field, solid temperature , and fluid fields. Since that the three fields are defined in different computational domains, three separate networks are trained. Aside from the differences in input and output dimensions, all other parameters remained consistent with those used in the decoupled scenario. The coupled datasets for experiments 1 and 2 consist of 10,000 and 5,000 samples, respectively, which is consistent with decoupled datasets. The model is evaluated using unseen coupling data during training.

The result is shown in Table 10, the accuracy of the model trained with decoupled data decreased by about 1 order of magnitude.

Table 10: Comparison of models trained on coupled and decoupled data.

|  | Coupled data model | Decoupled data model |
|---|---|---|
| Reaction-diffusion |  |  |
| $u$ | 0.00151 | 0.0141 |
| $v$ | 0.00185 | 0.0174 |
| Nuclear thermal coupling |  |  |
| neutron | 0.00512 | 0.0197 |
| solid | 0.00098 | 0.0287 |
| fluid | 0.00302 | 0.0391 |

# F ABLATION STUDY

## F.1 METHOD FOR CALCULATING THE ESTIMATED PHYSICAL FIELDS

We compare two methods for estimating physical fields: one using $z_i^e$ from Eq. 9 and the other using the current physical field $z_{i,s}$ with noise. As shown in Table 11, $z_i^e$ provides significantly better results than $z_{i,s}$, indicating that the estimate from $z_i^e$ is more accurate.

Table 11: Comparison of methods for estimating physical fields.

|  | $z_{i,s}$ | $z_i^e$ (Eq.9) |
|---|---|---|
| Reaction-diffusion |  |  |
| $u$ | 0.0525 | 0.0141 |
| $v$ | 0.0355 | 0.0174 |
| Nuclear thermal coupling |  |  |
| neutron | 0.0184 | 0.0197 |
| solid | 0.0913 | 0.0287 |
| fluid | 0.1000 | 0.0391 |
| Prismatic fuel element |  |  |
| $T$ | 0.0289 | 0.0076 |
| $\varepsilon$ | 0.0083 | 0.0194 |

## F.2 SELECTION OF HYPERPARAMETER $K$

This section examines how hyperparameter $K$ affects the predictive performance of multiphysics and multi-component problems in experiments 2 and 3. As shown in Tables 12 and 13, setting $K$ to 2

for multiphysics problems and $K$ to 3 for multi-component problems is adequate. The multiphysics algorithm updates physical fields at each diffusion time step, leading to faster convergence. In contrast, the multi-component problem relies on the field estimated in the previous time step for each diffusion iteration, resulting in slower convergence. Additionally, increasing $K$ further has a negligible effect on model performance.

Table 12: Hyperparameters of $K$ for multiphysics simulation.

| $K$ | neutron | solid | fluid |
|---|---|---|---|
| 1 | 0.0199 | 0.0304 | 0.0524 |
| **2** | 0.0206 | 0.0287 | 0.0391 |
| 3 | 0.0203 | 0.0288 | 0.0395 |

Table 13: Hyperparameters of $K$ for multi-component simulation.

| $K$ | $T$ | $\varepsilon$ |
|---|---|---|
| 1 | 0.00907 | 0.0236 |
| 2 | 0.00833 | 0.0222 |
| **3** | 0.00785 | 0.0206 |
| 4 | 0.00772 | 0.0207 |
| 5 | 0.00750 | 0.0203 |

### F.3 SELECTION OF HYPERPARAMETER OF $\lambda$

The hyperparameters $\lambda$ determine the weight of the current physical field, theoretically requiring a reliable estimate of 0 at the beginning and gradually increasing to 1 as diffusion progresses to provide better estimates of the current results in the later stage. We demonstrate this with experiment 3, which involves solving 64 components with slow convergence, making it sensitive to hyperparameter $\lambda$. We set the values to 0, 1, 0.5, and a linear increase. When $K$ is too large, result differences are minor, except when $\lambda$ is 1; thus, $K$ is set to 2. As shown in Table 14, employing a linearly increasing setting yields superior performance, which is consistent with the analysis.

Table 14: Hyperparameters of $\lambda$ for multi-component simulation.

| $\lambda$ | $T$ | $\varepsilon$ |
|---|---|---|
| 0 | 0.00878 | 0.0228 |
| 1 | 0.00913 | 0.0237 |
| 0.5 | 0.00895 | 0.0233 |
| **linear increase** | 0.00816 | 0.0217 |

## G THE DIFFERENCE BETWEEN TRAINING DATASET AND TESTING DATASET.

For multiphysics simulation, we train models for each physical process using decoupled data and combine them during testing to predict coupled solutions; for multi-component simulation, we train a model to predict individual component, then combine it during testing to predict the large structure composed of multiple components. To quantify the difference between the model's training and testing data, we calculate the Wasserstein distance (Feydy et al., 2019) between the training and validation data, as well as between the training and testing data, with the training and validation data originating from the same distribution. In addition, we also used the t-SNE (Van der Maaten & Hinton, 2008) algorithm to visualize this difference.

The results are presented in Table 15. In experiment 1, there is a significant difference between the training and testing data, as can be seen from Fig. 11, where only a small fraction of decoupled data points fall within the range of coupled data. In experiment 2, the difference between the training

and testing data for the neutron physics field is relatively small, likely due to the weak coupling effect of other physical processes on the neutron physics field. For the solid temperature field and fluid field, the difference between the training and testing data is also very pronounced, with almost no overlapping points in the Fig. 12. In experiment 3, since the range of training data has been expanded during data generation to cover as many potential scenarios of large structures as possible, the difference between the training and testing data is not as significant as in the multiphysics problem, and the testing data are also within the range of the training data, as shown in the Fig. 13.

Table 15: Wasserstein distance of datasets.

|  | Training and validation | Training and testing |
|---|---|---|
| Reaction-diffusion |  |  |
| $u$ | 0.343 | 52.7 |
| $v$ | 0.0435 | 20.3 |
| Nuclear thermal coupling |  |  |
| neutron | 42.4 | 31.3 |
| solid | 1.35 | 56.3 |
| fluid | 1.22 | 986 |
| Prismatic fuel element |  |  |
| $T$ | 0.233 | 9.05 |
| $\varepsilon$ | 0.625 | 12.5 |

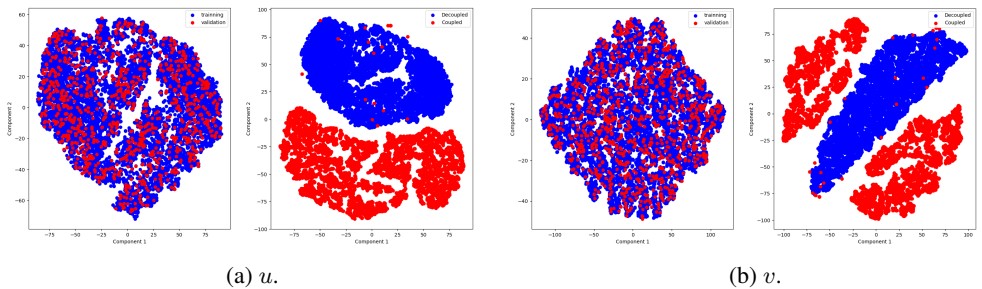

(a) $u$.      (b) $v$.

Figure 11: Visualization of experiment 1 Dataset.

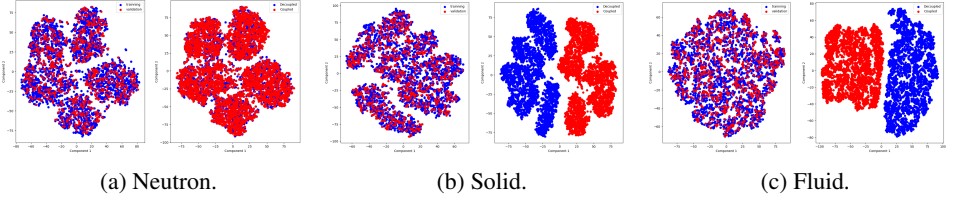

(a) Neutron.      (b) Solid.      (c) Fluid.

Figure 12: Visualization of experiment 2 Dataset.

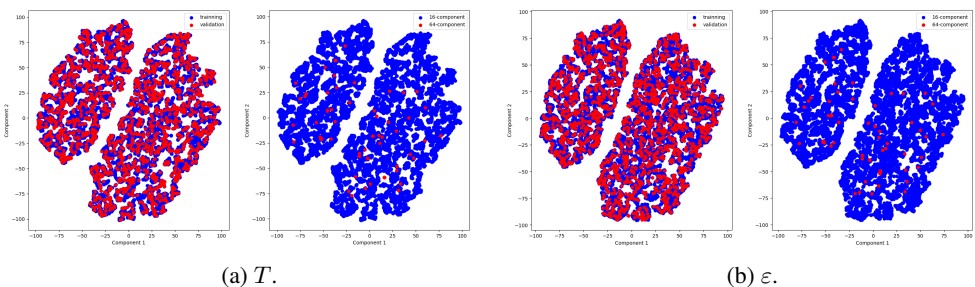

(a) $T$.      (b) $\varepsilon$.

Figure 13: Visualization of experiment 3 Dataset.

# H   SAMPLING ACCELERATION.

By employing the DDIM algorithm to expedite the sampling process of diffusion models, we have successfully enhanced the efficiency of model inference. The DDIM algorithm encompasses two parameters: the number of time steps $S$ and the parameter $\eta$, which controls the noise (Song et al., 2021):

$$\sigma_t = \eta \sqrt{\frac{1 - \overline{\alpha}_{t-1}}{1 - \overline{\alpha}_t}} \beta_t \tag{19}$$

where $\eta \in [0, 1]$. We conduct tests across various parameter combinations, including $S = 10, 25, 50$, and $\eta = 0, 0.5, 1$, with a particular focus on the model's performance in coupled and large structure prediction. These three experiments all use the most accurate model, which is: U-Net, U-Net, Transolver. Table 16 indicates that in experiment 1, the setting of $S = 25$ closely mirrors the results of $S = 50$, with $\eta$ having a relatively minor impact. Table 17 indicates that in experiment 2, the setting of $S = 25$ also approximates the outcome of $S = 50$ but is more sensitive to $\eta$, with $\eta = 1$ yielding the best performance. Table 18 indicates that in experiment 3, $S = 50$ is the optimal setting, and $\eta = 0$ provides the best results.

During the training of diffusion models, we uniformly set the number of time steps to 250. By employing accelerated sampling techniques, we achieved a 10-fold acceleration for multiphysics problems and a 5-fold acceleration for multi-component problem while ensuring the maintenance of predictive accuracy.

Table 16: Relative L2 norm of error on reaction-diffusion equation for DDIM sampling.

|  | $u$ | | $v$ | |
| method | decoupled | coupled | decoupled | coupled |
| --- | --- | --- | --- | --- |
| Original DDPM | 0.0119 | 0.0141 | 0.0046 | 0.0174 |
| $S = 10, \eta = 0$ | 0.0143 | 0.0170 | 0.0117 | 0.0215 |
| $S = 25, \eta = 0$ | 0.0123 | 0.0151 | 0.0082 | 0.0190 |
| $S = 50, \eta = 0$ | 0.0123 | 0.0147 | 0.0059 | 0.0179 |
| $S = 25, \eta = 0.5$ | 0.0123 | 0.0152 | 0.0082 | 0.0191 |
| $S = 25, \eta = 1$ | 0.0119 | 0.0151 | 0.0081 | 0.0192 |

Table 17: Relative L2 norm of prediction error on nuclear thermal coupling for DDIM sampling. The unit is $1 \times 10^{-2}$.

|  | neutron | | solid | | fluid | |
| method | decoupled | coupled | decoupled | coupled | decoupled | coupled |
| --- | --- | --- | --- | --- | --- | --- |
| Original DDPM | 0.487 | 1.97 | 0.108 | 2.87 | 0.303 | 3.91 |
| $S = 10, \eta = 1$ | 0.638 | 1.89 | 0.261 | 4.45 | 0.478 | 4.42 |
| $S = 25, \eta = 1$ | 0.552 | 2.03 | 0.142 | 3.64 | 0.343 | 4.08 |
| $S = 50, \eta = 1$ | 0.533 | 1.96 | 0.138 | 3.21 | 0.346 | 4.02 |
| $S = 25, \eta = 0.5$ | 2.82 | 2.78 | 0.793 | 5.28 | 0.970 | 4.70 |
| $S = 25, \eta = 0$ | 10.9 | 10.3 | 2.99 | 14.4 | 1.82 | 8.20 |

Table 18: Relative L2 norm of prediction error on prismatic fuel element experiment for DDIM sampling. The unit is $1 \times 10^{-2}$.

| method | single $T$ | single $\varepsilon$ | 16-component $T$ | 16-component $\varepsilon$ | 64-component $T$ | 64-component $\varepsilon$ |
|---|---|---|---|---|---|---|
| Original DDPM | 0.107 | 0.303 | 0.213 | 1.03 | 0.759 | 1.94 |
| $S = 10, \eta = 0$ | 0.207 | 0.425 | 1.69 | 3.81 | 1.89 | 4.13 |
| $S = 25, \eta = 0$ | 0.166 | 0.353 | 0.952 | 2.55 | 1.30 | 3.26 |
| $S = 50, \eta = 0$ | 0.158 | 0.337 | 0.669 | 1.87 | 0.865 | 2.31 |
| $S = 50, \eta = 0.5$ | 0.150 | 0.352 | 0.586 | 1.69 | 0.954 | 2.61 |
| $S = 50, \eta = 1$ | 0.130 | 0.322 | 0.553 | 1.62 | 1.05 | 2.80 |

# I EFFICIENCY ANALYSIS.

This section compares the computational efficiency of MultiSimDiff, surrogate model, and numerical programs. The time unit for each experiment is defined as the time required for a single neural network inference. These three experiments all use the most accurate model. Since the surrogate model and MultiSimDiff both use the same network architecture and have consistent network parameters, it is assumed that the time for a single inference using these two methods is equal. The numerical programs are run on the CPU and have all been optimized to the best parallel count.

Let the number of physical processes be denoted by $N$, the number of iterations for the surrogate model by $M$, the number of diffusion steps by $S$, and the number of outer loop iterations for the diffusion model by $K$. The computation time for the surrogate model is $M \times N$, while the diffusion model is $K \times S \times N$. The specific choices of $N, M, S, K$ for each experiment are presented in Table 19.

The results are presented in Table 20. In experiment 1, the problem is relatively simple, and the numerical algorithm achieves efficient solutions through explicit time stepping, while the introduction of MultiSimDiff actually reduces efficiency. However, in experiment 2, which addresses more complex problems, MultiSimDiff achieves a 29-fold acceleration compared to numerical programs. In experiment 3, comparing the results of 16 components with 64 components, it is observed that as the computational scale increases, the acceleration effect of MultiSimDiff becomes increasingly significant. Furthermore, when dealing with multi-component problems, the surrogate model requires iteration to ensure the convergence of solutions across all components. Due to the large number of components, the number of iterations needed significantly increases compared to multiphysics problems, resulting in higher efficiency for MultiSimDiff. In addition, we have only compared the efficiency of single computations for all experiments. When dealing with multiple problems simultaneously, the acceleration provided by MultiSimDiff will be even more pronounced due to the parallel nature of GPU computing.

In general, the more complex the problem, the more pronounced the acceleration effect of MultiSimDiff becomes. In fact, the problems in experiment 2 and experiment 3 have been simplified to a certain extent, and the actual situations are even more complex. Therefore, MultiSimDiff holds significant value in solving real-world complex engineering problems.

Table 19: Values of $K$, $N$, $M$, and $S$ for the three experiments.

| experiment | $N$ | $M$ | $S$ | $K$ |
|---|---|---|---|---|
| Reaction-diffusion | 2 | 27 | 25 | 2 |
| Nuclear thermal coupling | 3 | 21 | 25 | 2 |
| Prismatic fuel element (16-component) | 1 | 309 | 50 | 3 |
| Prismatic fuel element (64-component) | 1 | 324 | 50 | 3 |

Table 20: Comparison of running time.

| Experiment | Unit (s) | Numerical program | Surrogate model | MultiSimDiff | Speedup |
|---|---|---|---|---|---|
| Reaction-diffusion | 0.0115 | 6 | 54 | 100 | 0.064 |
| Nuclear thermal coupling | 0.0242 | 4368 | 63 | 150 | 29 |
| Prismatic fuel element (16-component) | 0.0067 | 834 | 309 | 150 | 5.6 |
| Prismatic fuel element (64-component) | 0.0256 | 6170 | 324 | 150 | 41 |

## J APPLICATION SCENARIOS FOR MULTI-COMPONENT SIMULATION.

In this section, we discuss the application scenarios of MultiSimDiff for multi-component simulation from both theoretical and practical perspectives.

From a theoretical perspective, in the derivation of Section 3.2, we make an assumption: the solution on a multi-component structure is an undirected graph that satisfies the local Markov property, meaning that any two non-adjacent variables are conditionally independent given all other variables. Using this property, we derived Eq. 14. We believe this assumption is applicable to most problems because physical fields are continuous in space, and the information exchange between any two points must be transmitted through the points in between. However, there is a class of problems to which current methods cannot be directly applied, which is the partial differential equation that requires determining eigenvalues:

$$\mathbf{M}\phi = \lambda\phi \tag{20}$$

Here $\mathbf{M}$ is the operator, $\lambda$ is the eigenvalue, $\phi$ is the physical field to be solved. The $\lambda$ varies with different systems, and the relationships we learn on small structures may not be applicable to large structures. Solutions to these problems may be similar to numerical algorithms, requiring the addition of an eigenvalue search process, which will be undertaken in future work.

From a practical implementation perspective, for a complex structure, it is necessary to clearly determine its basic components and the relationships between these components and their surrounding components, so that we can understand how the components are affected by their surrounding components. In addition, training data must encompass all possible scenarios that each component in a large structure might encounter, such as all possible boundary conditions and the relationships with surrounding components.

## K DATASETS DESCRIPTION.

This section provides a concise description of the datasets utilized in the three experiments, with their detailed backgrounds introduced in Appendix B, C, D. We outline the principal characteristics of these datasets and compare them with the standard scientific datasets PDEbench (Takamoto et al., 2022). Comparison is shown in Table 21, where $N_d$ is the spatial dimension, $N_f$ is the number of physical processes, and $N_c$ is the number of components. Table 21 only lists some of the datasets in PDEBench, but all of its datasets have $N_f$ and $N_c$ values of 1. The dataset of Experiment 1 in this paper exists in the benchmark, but Experiments 2 and 3 are completely new datasets.

Table 21: Datasets Description.

| PDE | $N_d$ | Time | Computational domain | $N_f$ | $N_c$ |
|---|---|---|---|---|---|
| Burgers' | 1 | yes | Line | 1 | 1 |
| compressible Navier-Stokes | 3 | yes | Cube | 1 | 1 |
| incompressible Navier-Stokes | 2 | yes | Rectangle | 1 | 1 |
| shallow-water | 2 | yes | Rectangle | 1 | 1 |
| **reaction-diffusion (Exp1)** | 1 | yes | Line | 1 | 1 |
| **heat conduction + neutron diffusion + incompressible Navier-Stokes (Exp2)** | 2 | yes | 3 Rectangle | 3 | 1 |
| **heat conduction + mechanics (Exp3)** | 2 | no | Irregular domain | 2 | 16 , 64 |

