# OpenReview forum: "Compositional Generative Multiphysics and Multi-component Simulation"
_ICLR.cc/2025/Conference — Submitted to ICLR 2025_

### Official Review · Reviewer_6eU9 · 2024-10-25

**Soundness:** 3
**Presentation:** 3
**Contribution:** 1
**Rating:** 5
**Confidence:** 4

**Summary:**

This paper addresses an important problem in the area of computational physics: namely, the composition (coupling) of multi-physics and multi-component simulations. The authors present a 'Bayesian approach' to model composition, with Eq 6 being the 'foundation' of their proposed method.

The reviewer agrees with the importance of the problem and with the general (proposed) approach.  The reviewer, however, does not find the literature review to be sufficiently broad as the proposed method (as given) has been proposed in the computational mechanics literature they use as motivation.

**Strengths:**

The main strength of this paper is the use of a Bayesian approach to allow simulation scientists to 'decouple' their solvers by characterizing them in a probabilistic way.

**Weaknesses:**

The major weakness of the paper is that it is not clear that the method is novel.  For those who are embedded in the work of D.A. Knoll and D.E. Keyes (paper referenced by the authors -- good paper), George El Haber, Jonathan Viquerat, Aurelien Larcher, David Ryckelynck, Jose Alves, Aakash Patil, and Elie Hachem (JCP paper referenced), etc.,  --- these people (e.g., David Keyes) would probably start a history lesson with pointing out the seminar paper of Kennedy and O'Hagan (Bayesian calibration of computer models, Jan 2002) as the starting point of an entire class of methods on using the Bayesian approach for coupling, uncertainty estimation, etc.  With the Kennedy and O'Hagan paper as a starting point, you get things like:

https://amses-journal.springeropen.com/articles/10.1186/s40323-022-00237-5
(and lots of Wolfgang Wall's work)

http://mcubed.mit.edu/files/public/RT3/2016__Allaire__Quantifying_Model_Discrepancy_in_coupled_multi-physics_systems.pdf

https://www.sciencedirect.com/science/article/pii/S0021999119304206

and then particular people like Karen Willcox (UT-Austin), Youssef Marzouk (MIT), etc. and their use of Bayesian methods for "all kinds of things."

Given that there is a rich history of these methods within the journals referenced by the authors and given that it is difficult to evaluate the novelty of the statements against this 20 year history, the reviewer (at this time) cannot recommend the paper for acceptance.

**Questions:**

What is the novelty of the method in comparison to the papers mentioned above and more broadly the papers/journals in which these papers reside?  The papers mentioned below are not necessarily the seminal papers, but what one gets by googling with keywords associated with the topic and the journals mentioned by the authors.

Specifically, in terms of addressing the weaknesses:

+ How does the author's compositional diffusion model approach compare to the Bayesian calibration methods of Kennedy & O'Hagan and subsequent work that builds on this (of which the reviewer has given some, but which is a vast area)?

+ Whether and how the author's method of learning conditional energy functions and composing them differs from existing Bayesian coupling approaches?  The Bayesian approach as presented seems consistent with what a practitioner might do for uncertainty quantification, but does not replace the weak and strong coupling methods mentioned in the early part of the paper (which is interested in specific instances, not probabilistic statements).

The author is asking If the focus on using decoupled training data to predict coupled solutions, and small structure data to predict large structures, represents a novel contribution.

**Details Of Ethics Concerns:**

No ethics concerns.

---

> ### Author Response · Authors · 2024-11-22
> **Official Response to Reviewer 6eU9 (1)**
>
> We appreciate you pointing out the research field of Bayesian calibration, which could be beneficial for our future research. We have read through all the articles you have recommended, but upon thorough analysis, we find that these Bayesian-based methods primarily focus on analyzing **well-established** multi-physics simulation systems, including uncertainty analysis, error calibration, and decoupling approximations of coupled systems.
>
> However, the motivation behind our method when applied to multiphysics simulation is that constructing coupled programs for complex problems can be exceedingly **intricate** and computationally inefficient. Our aim is to directly predict coupled solutions through decoupled programs, thereby eliminating the need for coupled program construction. For multi-component simulation, directly simulating the overall structure requires high computational cost and may encounter difficulties in convergence due to the increase in degrees of freedom. Our aim is to predict the large structure through small structure data. In the fields of engineering and scientific research, virtually all physicochemical processes involve the combined effects of multiple physical processes. Multi-component  is common in large complex systems such as aircraft, buildings, and nuclear power plants, as well as in systems with highly intricate internal structures like chips and batteries. Rapid and accurate solutions to multiphysics and multi-component problems can lead to a better understanding of the physical behavior of systems, thereby enhancing their economic and reliability performance.
>
> Thus, we believe our method is fundamentally **different** from the work described in these articles. We use decoupled training data to predict coupled solutions, and small structure data to predict large structures. This method shows great significance in science and engineering, representing a novel contribution.
>
> The detailed analysis of the articles you mentioned is in the next response.

---

> ### Author Response · Authors · 2024-11-22
> **Official Response to Reviewer 6eU9 (2)**
>
> For reference [1], this article employs the JFNK method to solve large systems of equations, which is a full-coupling strategy and a numerical technique for addressing multiphysics coupling problems. However, the full-coupling approach is not suitable for all scenarios, particularly when two physical processes require different numerical methods for their solutions. In addition, the cost of JFNK for solving complex problems is high.
>
> For reference [2], as mentioned in the manuscript, this article establishes a surrogate model for the temperature field and then integrates it with other CFD solvers to accelerate multiphysics simulation. However the issue discussed is only a unidirectional coupling problem, while bidirectional coupling may introduce additional challenges. In contrast, our algorithm is not limited in this regard, and Experiment 2 encompasses a variety of scenarios that multiphysics simulation might encounter.
>
> For reference [3,4,5,6], these articles are actually focused on the quantification of uncertainty in multiple numerical programs, which is entirely different from our work. The motivation is that numerical programs may have inaccurate internal parameters in new scenarios. To address this issue, the article calibrates the internal parameters of the program with new experimental data to ensure that numerical predictions match observed data.
>
> As you mentioned, researcher Karen Willcox from UT-Austin is a highly influential scholar in her field. Her research focuses on model reduction, multifidelity methods, digital twins, and uncertainty quantification. The theme closest to this article is model reduction, which can also be understood as the use of surrogate models. In our manuscript, we use the surrogate model as a baseline for comparison and our method demonstrates superior performance. Researcher Youssef Marzouk from MIT is also a highly influential scholar in his field. His research focuses on uncertainty quantification, inverse problems, and Bayesian statistics. He has done extensive work using Bayesian methods in inverse problems and uncertainty analysis. In the field of multiphysics, he uses Bayesian methods to identify the most critical factors in coupled multi-physics systems. However, our research is actually concerned with **forward** simulation problems.
>
> We hope the explanation above has answered all of the questions.
>
> [1] Knoll, Dana A., and David E. Keyes. "Jacobian-free Newton–Krylov methods: a survey of approaches and applications." Journal of Computational Physics 193.2 (2004): 357-397.
>
> [2] El Haber, George, et al. "Deep learning model to assist multiphysics conjugate problems." Physics of Fluids 34.1 (2022).
>
> [3] Kennedy, Marc C., and Anthony O'Hagan. "Bayesian calibration of computer models." Journal of the Royal Statistical Society: Series B (Statistical Methodology) 63.3 (2001): 425-464.
>
> [4] Friedman, Samuel, and Douglas Allaire. "Quantifying Model Discrepancy in Coupled Multi-Physics Systems." International Design Engineering Technical Conferences and Computers and Information in Engineering Conference. Vol. 50077. American Society of Mechanical Engineers, 2016.
>
> [5] Willmann, Harald, et al. "Bayesian calibration of coupled computational mechanics models under uncertainty based on interface deformation." Advanced modeling and simulation in engineering sciences 9.1 (2022): 24.
>
> [6] Subramanian, Abhinav, and Sankaran Mahadevan. "Error estimation in coupled multi-physics models." Journal of Computational Physics 395 (2019): 19-37.

---

> > ### Author Response · Authors · 2024-11-25
> > **A gentle reminder: please respond to our rebuttal**
> >
> > Dear Reviewer 6eU9,
> >
> > Thank you for your time and effort in reviewing our work. We have carefully considered your detailed comments and questions, and we have tried to address all your concerns accordingly.
> >
> > As the deadline for author-reviewer discussions is approaching, could you please go over our responses? If you find our responses satisfactory, we hope you could consider adjusting your initial rating. Please feel free to share any additional comments you may have.
> >
> > Thank you!
> >
> > Authors

---

> > > ### Comment · Area_Chair_N5rq · 2024-11-26
> > >
> > > Dear reviewer,
> > >
> > > Please make to sure to read, at least acknowledge, and possibly further discuss the authors' responses to your comments. Update or maintain your score as you see fit.
> > >
> > > The AC.

---

> ### Author Response · Authors · 2024-12-03
> **A gentle reminder: please respond to our rebuttal**
>
> Dear Reviewer 6eU9,
>
> Thank you for your time and effort in reviewing our work. We have carefully considered your detailed comments and questions, and we have tried to address all your concerns accordingly.
>
> As the deadline for author-reviewer discussions is approaching, could you please go over our responses? If you find our responses satisfactory, we hope you could consider adjusting your initial rating. Please feel free to share any additional comments you may have.
>
> Thank you!
>
> Authors

---

### Official Review · Reviewer_1Pcu · 2024-10-30

**Soundness:** 2
**Presentation:** 3
**Contribution:** 2
**Rating:** 5
**Confidence:** 3

**Summary:**

The paper presents MultiSimDiff, a novel generative approach for Multiphysics and multi-component simulations, using diffusion models to overcome the limitations of surrogate models. MultiSimDiff framework can be seamlessly integrated with existing backbone architectures, modeling the conditional probability of each physical field or component within a system. By training on decoupled data, it can generate joint solutions through reverse diffusion. This method demonstrates high accuracy, including reaction-diffusion, nuclear thermal coupling and prismatic fuel elements.

**Strengths:**

- The application of novel machine learning techniques, (the use of diffusion models in scientific domains is relatively new), particularly for multiphysics and multi-component simulations.
- Trained on small, decoupled datasets, this method can provide solutions for extended, unseen data composed of smaller components, showing great potential for applications across various scientific and engineering fields.
- The benchmarks are somewhat new and practical, beyond traditional toy PDE benchmarks.

**Weaknesses:**

- The multi-level “for loops” can cause a significant computational bottleneck, limiting the practical applicability of the method. While the author briefly acknowledged this limitation, they did not provide any metrics on the method’s computational efficiency, particularly in comparison to surrogates, which are a key consideration for physical simulations and their surrogates.
- Although the authors noted the application of various compositional generative models in scientific domains in their related works, no compositional baselines were compared in the experiments.
- In some experiments, the combination of “surrogate+x” outperformed the proposed methods.
- Although interesting, it remains unclear how models trained on small structures can effectively extrapolate to larger structures. For instance, a model trained on a single pendulum cannot easily predict the behavior of a double pendulum, as the interactions within coupled systems add complexity beyond a simple combination. Could you clarify the difference between this scenario and the benchmarks used in paper, or at least explain if this method can be applied to this scenario?

**Questions:**

See above.

**Details Of Ethics Concerns:**

See above.

---

> ### Author Response · Authors · 2024-11-22
> **Official Response to Reviewer 1Pcu (1)**
>
> We thank the reviewer for the constructive review, and are glad that the reviewer recognizes the novelty and generality of our method, the practicality of the dataset. In the following, we address the points the reviewer raised about, and answer the questions.
>
>
> >Re1: The multi-level “for loops” can cause a significant computational  bottleneck, limiting the practical applicability of the method. While  the author briefly acknowledged this limitation, they did not provide  any metrics on the method’s computational efficiency, particularly in  comparison to surrogates, which are a key consideration for physical  simulations and their surrogates.
>
> Answer: We greatly appreciate your feedback. Efficiency is indeed very important. Firstly, for the outmost loop that go over $k=1,2,...K$, we find that typically $K=2$ or $3$ is enough, and set $K=2$ for multiphysics simulation and $K=3$ for multi-component simulation. The second loop is for the denoising steps for diffusion models. To enhance efficiency, we have implemented DDIM for accelerated sampling in appendix H and conducted a comparative analysis of efficiency in appendix I. In the more complex Experiments 2 and 3, our method achieves **29 times** and **41 times** acceleration compared to numerical programs. When compared to the surrogate model, our algorithm takes approximately 2-3 times longer to run for multiphysics problems. For multi-component problems, our algorithm is more efficient, with a running time of about half that of the surrogate model. The results are shown in Table 20. Additionally, we would like to clarify that the use of multi-level  "for loops" is primarily aimed at enhancing accuracy. In scenarios where efficiency is prioritized, **the outmost loop can be omitted**, and this adjustment does not significantly impact accuracy as shown in appendix F where we perform a hyperparameter study of $K$. Table 20 is also provided as follows, the time unit for each experiment is defined as the time required for a single neural network inference.
>
> | Experiment                            | Unit (s) | Numerical program | Surrogate model | model | Speedup |
> | ------------------------------------- | -------- | ----------------- | --------------- | ----- | ------- |
> | Exp 1: Reaction-diffusion                    | 0.0115   | 6                 | 54              | 100   | 0.064   |
> | Exp 2: Nuclear thermal coupling              | 0.0242   | 4368              | 63              | 150   | 29      |
> | Exp 3: Prismatic fuel element (16-component) | 0.0067   | 834               | 309             | 150   | 5.6     |
> | Exp 3: Prismatic fuel element (64-component) | 0.0256   | 6170              | 324             | 150   | 41      |
>
> >Re2: Although the authors noted the application of various compositional  generative models in scientific domains in their related works, no  compositional baselines were compared in the experiments.
>
> Answer: Thank you for your comment. In fact, before starting on this work, we had considered using existing compositional generation methods. However, we found that these methods are not applicable to our problem. The most relevant studies to our issue are those by Du et al. [1] in the field of image generation and Wu et al. [2] who further extended the concept to inverse design problems in science. Their focus is on how **a single object is influenced by multiple factors**, such as generating images that meet various requirements in image generation or predicting the fluid fields and enhancing the lift-to-drag ratio under the influence of two wings in inverse design. However, our problem involves **multiple objects**, such as multiple physical processes and components, requiring the capture of interactions between these processes or components. Therefore, their methods are not suitable for our scenario.
>
> [1]Du, Yilun, et al. "Reduce, reuse, recycle: Compositional generation with energy-based diffusion models and mcmc." ICML, 2023.
>
> [2]Wu, Tailin, et al. "Compositional Generative Inverse Design." ICLR, 2024.

---

> > ### Author Response · Authors · 2024-11-22
> > **Official Response to Reviewer 1Pcu (2)**
> >
> > >Re3: In some experiments, the combination of “surrogate+x” outperformed the proposed methods. .
> >
> > Answer: Firstly, we would like to clarify that there is a significant difference between the training data and the test data in our experiments, and our primary focus is on the model's performance on the test data. When dealing with multiphysics problems, we pay special attention to the model's predictive capabilities on coupled data; while in multi-component problems, we focus on the accuracy of the model's predictions for larger structures than in training. In the original submission, our algorithm performed optimally in experiment 2 and experiment 3. However, in experiment 1, when the network architecture is FNO, the surrogate model shows superior performance in predicting the physical quantity "v". We must apologize for the error that occurred in the previous code, during the training of the FNO-2D diffusion model, where the diffusion time step was not correctly inputted, leading to significant prediction errors in our method. Now Table 1 has been revised in the latest manuscript, and **our algorithm demonstrates optimal performance in all experiments**. Part of Table 1 is also provided as follows:
> >
> > | Method                        | u (decoupled) | u (coupled) | v (decoupled) | v (coupled) |
> > | ----------------------------- | ------------- | ----------- | ------------- | ----------- |
> > | surrogate + FNO               | 0.0669        | 0.0600      | 0.0080        | 0.0320      |
> > | **MultiSimDiff (ours) + FNO** | 0.0270        | **0.0290**  | 0.0102        | **0.0264**  |

---

> > > ### Author Response · Authors · 2024-11-22
> > > **Official Response to Reviewer 1Pcu (3)**
> > >
> > > >Re4: Although interesting, it remains unclear how models trained on small  structures can effectively extrapolate to larger structures. For  instance, a model trained on a single pendulum cannot easily predict the  behavior of a double pendulum, as the interactions within coupled  systems add complexity beyond a simple combination. Could you clarify  the difference between this scenario and the benchmarks used in paper,  or at least explain if this method can be applied to this scenario?
> > >
> > > Answer: Thanks for your insightful feedback. That is a very good question. We have added a description of the application scenarios in the appendix.
> > >
> > > Firstly, from a theoretical standpoint, in the derivation of Section 3.2, we made an assumption: we consider the solution on a multi-component structure to be an undirected graph that satisfies the local Markov property, meaning that any two non-adjacent variables are conditionally independent given all other variables. Using this property, we derived Equation 14. We believe this assumption is applicable to most problems because physical fields are **continuous** in space, and the information exchange between any two points must be transmitted through the points in between. Essentially, our method learns **local conditional probability distributions**. In the inference time, these local conditional probabilistic models can compose to generate **globally unseen distributions**, although locally they are **in distribution**. This is the core reason why our method can extrapolate to larger structures in inference.
> > >
> > >
> > > On the other hand, we find that there is a class of problems to which current methods cannot be directly applied, which is the partial differential equation that requires determining eigenvalues. The eigenvalues of these problems vary with different systems, and the relationships we learn on small structures may not be applicable to large structures. Solutions to these problems may be similar to numerical algorithms, requiring the addition of an eigenvalue search process, which will be undertaken in future work.
> > >
> > > From a practical implementation perspective, for a complex structure, it is necessary to clearly determine its **basic components** and the relationships between these components and their surrounding components, so that we can understand how the components are affected by their surrounding components. In addition, training data must encompass all possible scenarios that each component in a large structure might encounter, such as all possible boundary conditions and the relationships with surrounding components, so that in the inference, they are **locally in distribution**.
> > >
> > > Lastly, regarding the double pendulum problem you mentioned, we indeed cannot infer the behavior of a double pendulum from the data of a single pendulum. In a double pendulum system, the balls exist in two states: one end in contact with wall and the other with a ball; or one end with a ball and the other empty. In contrast, a single pendulum system only has one end in contact with wall and the other empty. Therefore, it is not possible to obtain the solution for a double pendulum by combining data from a single pendulum. For the more general n-pendulum system, the balls exist in three states: one end in contact with the wall and the other with a ball; one end with a ball and the other empty; both ends with balls. These states are all present in a triple pendulum system, so perhaps we can simulate an n-pendulum system with data from a triple pendulum system. However, in practice, the chaotic nature of the system may amplify the errors between our method and the ground truth, leading to failure. In addition, solving a triple pendulum system is already quite complex, making verification very difficult.

---

> > > > ### Author Response · Authors · 2024-11-25
> > > > **A gentle reminder: please respond to our rebuttal**
> > > >
> > > > Dear Reviewer 1Pcu,
> > > >
> > > > Thank you for your time and effort in reviewing our work. We have carefully considered your detailed comments and questions, and we have tried to address all your concerns accordingly.
> > > >
> > > > As the deadline for author-reviewer discussions is approaching, could you please go over our responses? If you find our responses satisfactory, we hope you could consider adjusting your initial rating. Please feel free to share any additional comments you may have.
> > > >
> > > > Thank you!
> > > >
> > > > Authors

---

> > > > > ### Comment · Area_Chair_N5rq · 2024-11-26
> > > > >
> > > > > Dear reviewer,
> > > > >
> > > > > Please make to sure to read, at least acknowledge, and possibly further discuss the authors' responses to your comments. Update or maintain your score as you see fit.
> > > > >
> > > > > The AC.

---

> ### Author Response · Authors · 2024-12-03
> **A gentle reminder: please respond to our rebuttal**
>
> Dear Reviewer 1Pcu,
>
> Thank you for your time and effort in reviewing our work. We have carefully considered your detailed comments and questions, and we have tried to address all your concerns accordingly.
>
> As the deadline for author-reviewer discussions is approaching, could you please go over our responses? If you find our responses satisfactory, we hope you could consider adjusting your initial rating. Please feel free to share any additional comments you may have.
>
> Thank you!
>
> Authors

---

### Official Review · Reviewer_Fscm · 2024-11-01

**Soundness:** 3
**Presentation:** 3
**Contribution:** 4
**Rating:** 8
**Confidence:** 4

**Summary:**

Article summary: This article is about the application of diffusion model in engineering model (mainly thermal and material science). Because engineering problems often involve multiple physical processes (that is, multiple complex processes are involved when modeling), this article establishes a multi-process model to deal with different physical problems. (Mostly reflected in reaction diffusion and nuclear thermal coupling problems)

**Strengths:**

Advantages:

1. The starting point of this article is very novel, hoping to use limited data to deal with more physical problems at the same time. Especially a mechanical structure and thermal problem, and the performance is very good from the results of the article. (Especially page 18)

2. The adaptability of the ML model to the actual parameters of the engineering problem makes me feel that the design of the entire model is justified, and the motivation is clear and credible. (That is, the content in the appendix, Tables 10-13)

3. Strict model comparison, such as using consistent hyperparameters and settings, and having different parameter designs in different engineering problems (such as porous/multi-part materials).

4. The superiority of the model, from the perspective of energy (density probability form), some of the results obtained are indeed very good

**Weaknesses:**

Disadvantages:

1. In the comparative experiment, is the surrogate model too simple? Or can you compare your model with a more complex model? Or explain the current popularity of the surrogate model.

2. I would like to know whether the model you designed has other innovations in structure and implementation compared with the diffusion model, in addition to the differences in parameter settings and application issues. I feel that the model is not deep enough, judging from the algorithms shown in the first six pages. (I will consider this again)

3. I also found that your experiments often combine your model with other NNs (FNO, etc.). Why don't you use your model to implement it independently? Because I am worried that NNs such as FNO will additionally correct the errors of your model (if they occur).

**Questions:**

There are some unnecessary blank lines in the article that can be corrected. I will be happy to improve my score in subsequent discussions.

**Details Of Ethics Concerns:**

None. This is original work and there are no ethical issues

---

> ### Author Response · Authors · 2024-11-22
> **Official Response to Reviewer Fscm**
>
> We thank the reviewer for the constructive review, and are glad that the reviewer recognizes the novelty of our method and the credibility of the experimental results. In the following, we address the points the reviewer raised about, and answer the questions.
>
>
> >Re1: In the comparative experiment, is the surrogate model too simple?  Or can you compare your model with a more complex model? Or explain the  current popularity of the surrogate model.
>
> Answer: Firstly, we would like to clarify that in each experiment, for each network architecture, we have concurrently trained **both** diffusion models and surrogate models. Our primary objective is to compare the differences between our approach of multiphysics and multi-component simulation by composing diffusion models, and the conventional approach of learning surrogate models, **rather** than the differences between the network architectures themselves. In fact, both our approach and the conventional surrogate models approach can use suitable network architectures. Secondly, for the same network structure, both diffusion models and surrogate models employ the same network hyperparameters and training methods, thus their complexity is nearly identical. Furthermore, the FNO and U-Net network structures used in this manuscript are common baseline models in the neural PDE tasks, as demonstrated in studies by Georg Kohl et al. [1], Wang et al. [2], and Xiong et al. [3], among others. The Transolver used in Experiment 3, although newly proposed, is essentially a Transformer, an architecture that has been widely applied to a variety of tasks.
>
>
> [1] Kohl, Georg, Liwei Chen, and Nils Thuerey. "Benchmarking autoregressive conditional diffusion models for turbulent flow simulation." ICML 2024 AI for Science Workshop. 2024.
>
> [2] Wang, Haixin, et al. "Recent advances on machine learning for computational fluid dynamics: A survey." arXiv preprint arXiv:2408.12171 (2024).
>
> [3] Xiong, Wei, et al. "Koopman neural operator as a mesh-free solver of non-linear partial differential equations." Journal of Computational Physics (2024): 113194.
>
>
> >Re2: I would like to know whether the model you designed has other  innovations in structure and implementation compared with the diffusion  model, in addition to the differences in parameter settings and  application issues. I feel that the model is not deep enough, judging  from the algorithms shown in the first six pages.
>
> Answer: Our primary contribution lies in addressing multiphysics and multi-component simulation problems from a probabilistic generation perspective. In the fields of engineering and scientific research, virtually all physicochemical processes involve the combined effects of multiple physical processes. Multi-component  is common in large complex systems such as aircraft, buildings, and nuclear power plants, as well as in systems with highly intricate internal structures like chips and batteries. Rapid and accurate solutions to multiphysics and multi-component problems can lead to a better understanding of the physical behavior of systems, thereby enhancing their economic and reliability performance. We have theoretically deduced the feasibility of our approach and validated its effectiveness with examples from engineering. We employ diffusion models to tackle these issues, making certain improvements to tailor them to our specific problems. In theory, other types of generative models, such as flow-based models, could also be utilized.
>
>
> >Re3: I also found that your experiments often combine your model with  other NNs (FNO, etc.). Why don't you use your model to implement it  independently? Because I am worried that NNs such as FNO will  additionally correct the errors of your model (if they occur).
>
> Answer: We would like to clarify that our contribution does not lie in the proposal of a new network architecture, but rather in the application of diffusion models from a probabilistic generation perspective to solve multiphysics and multi-component simulation problems. In addition, the diffusion model is a learning paradigm that can be implemented using various network architectures such as U-Net, FNO, and transformer. Our approach is theoretically versatile and can be applied to various network architectures, including U-Net and FNO, as well as various generative models such as diffusion models and flow-base models.
>
>
>
> >Re4: There  are some unnecessary blank lines in the article that can be corrected.
>
> Answer: Thank you for pointing out this issue. We have adjusted the layout to reduce the presence of blank lines.

---

> > ### Author Response · Authors · 2024-11-25
> > **A gentle reminder: please respond to our rebuttal**
> >
> > Dear Reviewer Fscm,
> >
> > Thank you for your time and effort in reviewing our work. We have carefully considered your detailed comments and questions, and we have tried to address all your concerns accordingly.
> >
> > As the deadline for author-reviewer discussions is approaching, could you please go over our responses? If you find our responses satisfactory, we hope you could consider adjusting your initial rating. Please feel free to share any additional comments you may have.
> >
> > Thank you!
> >
> > Authors

---

> > > ### Comment · Area_Chair_N5rq · 2024-11-26
> > >
> > > Dear reviewer,
> > >
> > > Please make to sure to read, at least acknowledge, and possibly further discuss the authors' responses to your comments. Update or maintain your score as you see fit.
> > >
> > > The AC.

---

> > > ### Comment · Reviewer_Fscm · 2024-12-01
> > >
> > > Your answer has cleared my doubts to a certain extent. I am very grateful and I have improved my score.

---

> > > > ### Author Response · Authors · 2024-12-02
> > > > **Official Comment by Authors**
> > > >
> > > > Thank you for your constructive feedback and for raising our manuscript's score. Your insights have been invaluable in enhancing the quality of our work.

---

### Official Review · Reviewer_HRRS · 2024-11-03

**Soundness:** 3
**Presentation:** 3
**Contribution:** 3
**Rating:** 5
**Confidence:** 3

**Summary:**

The author proposed compositional Multiphysics and Multi-component Simulation with Diffusion models (MultiSimDiff) to overcome the diffculty of solving complex systems

**Strengths:**

The paper is good with the originality and create a novel approach on multiphysics simulation

**Weaknesses:**

The algorithm lacks clarity, and the model structure is not sufficiently detailed.

The iterative nature of MultiSimDiff, especially in multiphysics simulations, requires multiple diffusion steps for each field, which may lead to slow inference times. This constraint limits its practicality for scenarios requiring rapid predictions. While the authors recognize this issue and propose exploring faster sampling methods in future work, an initial investigation into such techniques within this paper could enhance its contribution.

**Questions:**

1. For algorithm 1, it is identical to the alogrithm 2 in the DDPM paper [1]. And also the way identify $z_i$ is confusion when apply the third for loop.
2. How time in involved in the alogrithm as you are doing time dependent system




[1] Jonathan Ho, Ajay Jain, and Pieter Abbeel. Denoising diffusion probabilistic models. In
H. Larochelle, M. Ranzato, R. Hadsell, M.F. Balcan, and H. Lin (eds.), Advances in Neural Information Processing Systems, volume 33, pp. 6840–6851. Curran Associates, Inc.,
2020. URL https://proceedings.neurips.cc/paper_files/paper/2020/
file/4c5bcfec8584af0d967f1ab10179ca4b-Paper.pdf.

---

> ### Author Response · Authors · 2024-11-22
> **Official Response to Reviewer HRRS**
>
> We thank the reviewer for the constructive review, and are glad that the reviewer recognizes the novelty of our method. In the following, we address the points the reviewer raised about, and answer the questions.
>
>
> >Re1: The algorithm lacks clarity, and the model structure is not sufficiently detailed.
>
> Answer: Our algorithm builds on top of diffusion models to make it suitable for multiphysics and multi-component simulations. It can combine with any network architecture as its denoising network, so long as the network architecture fits the data structure. The network architecture we have chosen includes widely used U-Net and FNO in their original form, and Geo-FNO and Transolver in the multi-component experiment. The parameter details of these network architectures are provided in the Appendix B, C, and D in the manuscript.
>
> >Re2: The iterative nature of MultiSimDiff, especially in multiphysics  simulations, requires multiple diffusion steps for each field, which may  lead to slow inference times. This constraint limits its practicality  for scenarios requiring rapid predictions. While the authors recognize  this issue and propose exploring faster sampling methods in future work,  an initial investigation into such techniques within this paper could  enhance its contribution.
>
> Answer: Thank you for the suggestions. We have employed DDIM for accelerated sampling, the result is shown in appendx H. Compared to the original DDPM, it can achieve a 10-fold and 5-fold acceleration for multi-physics and multi-component problems, respectively, while maintaining accuracy. Additionally, we compared it with numerical programs in Table 20. In Experiment 2 and Experiment 3, we achieved accelerations of **29 and 41 times**, respectively. The more complex the problem, the higher the efficiency of our algorithm compared to numerical programs. Table 20 is also provided as follows, the time unit for each experiment is defined as the time required for a single neural network inference.
>
> | Experiment                            | Unit (s) | Numerical program | Surrogate model | model | Speedup |
> | ------------------------------------- | -------- | ----------------- | --------------- | ----- | ------- |
> | Exp 1: Reaction-diffusion                    | 0.0115   | 6                 | 54              | 100   | 0.064   |
> | Exp 2: Nuclear thermal coupling              | 0.0242   | 4368              | 63              | 150   | 29      |
> | Exp 3: Prismatic fuel element (16-component) | 0.0067   | 834               | 309             | 150   | 5.6     |
> | Exp 3: Prismatic fuel element (64-component) | 0.0256   | 6170              | 324             | 150   | 41      |
>
> >Re3: For algorithm 1, it is identical to the alogrithm 2 in the DDPM  paper [1]. And also the way identify $z_i$ is confusion when apply the  third for loop.
>
> Answer: Thank you for the question. Our primary contribution lies in addressing multiphysics and multi-component simulation problems from a probabilistic generation perspective instead of the improvement of diffusion model. We have theoretically deduced the feasibility of our approach and validated its effectiveness with examples from engineering. We employ diffusion models to tackle these issues, building on top of them to address our specific problems. In theory, other types of generative models, such as flow-based models, could also be utilized.
>
> As for $z_i$, the $z_i$ in the third loop of Algorithm 1 represents the i-th physical fields with noise at the current diffusion step. The objective is to denoise it to obtain the desired coupled solution, which is similar to DDPM's denoising of images.
>
> >Re4: How time in involved in the alogrithm as you are doing time dependent system?
>
> Answer: Both Experiment 1 and Experiment 2 deal with time-dependent systems. In these experiments, we treat time as an additional dimension, and **generate the state trajectory across all time simultaneously**. Consequently, for Experiment 1, which is a one-dimensional spatial-temporal problem, we employ a two-dimensional network; for Experiment 2, which is a two-dimensional spatial-temporal problem, we utilize a three-dimensional network.

---

> > ### Author Response · Authors · 2024-11-25
> > **A gentle reminder: please respond to our rebuttal**
> >
> > Dear Reviewer HRRS,
> >
> > Thank you for your time and effort in reviewing our work. We have carefully considered your detailed comments and questions, and we have tried to address all your concerns accordingly.
> >
> > As the deadline for author-reviewer discussions is approaching, could you please go over our responses? If you find our responses satisfactory, we hope you could consider adjusting your initial rating. Please feel free to share any additional comments you may have.
> >
> > Thank you!
> >
> > Authors

---

> > > ### Comment · Area_Chair_N5rq · 2024-11-26
> > >
> > > Dear reviewer,
> > >
> > > Please make to sure to read, at least acknowledge, and possibly further discuss the authors' responses to your comments. Update or maintain your score as you see fit.
> > >
> > > The AC.

---

> > > ### Comment · Reviewer_HRRS · 2024-12-02
> > >
> > > Dear Authors,
> > >
> > > Thank you for addressing the comments and suggestions raised during the review process. I appreciate your detailed responses and the revisions made to improve the manuscript. Upon careful evaluation of the updated submission, I have decided to retain the original marks assigned during the review. Thank you for your effort and engagement throughout the review process.

---

### Official Review · Reviewer_9cxV · 2024-11-04

**Soundness:** 2
**Presentation:** 4
**Contribution:** 3
**Rating:** 5
**Confidence:** 3

**Summary:**

The paper proposes a diffusion-based model for simulating multiphysics, multi-component systems. The model learns the conditional distributions of each field/component given the others. During inference, the model iteratively denoises each field/component in a manner similar to diffusion models. The model is applied on reaction-diffusion, nuclear thermal and prismatic fuel element datasets.

**Strengths:**

The paper is well-written and clear, presenting a novel combination of ideas by proposing to use diffusion models for simulating multiphysics, multi-component systems.

**Weaknesses:**

- the main claim of the papers (2) and (3) are questionable. See the below two points
- claim (2): There are several datasets in the literature that can be described as simulations of multiphysics, multi-component, such as PDEBench [Takamoto et al., 2022]. The authors should better precise why the new datasets they propose is a contribution to the literature.
- claim (3): More importantly, based on the results, it is difficult to determine whether the proposed method is advantageous for other multiphysics, multi-component systems, given the fact that it may not be competitive in terms of computational cost (see below question)

Minor remarks.
- l102. "a process we have mathematically proven", I don't think a "process" qualifies as being able to be "mathematically proven". You should precise what is "mathematically proven"
- l107. "This reverse diffusion process is also mathematically validated", same remark
- l137. "there do not exist utilized machine learning methods for multi-component simulation". I don't really understand the novelty since, technically, UNets [Ronneberger et al. 2015], FNO [Li et al. 2022], Transformer [Mccabe et al. 2023], diffusion [Kohl et al. 2024] models are already used on multi-component simulations.
- l251. Paragraph 3.2 is confusing. In particular, there can be confusion between "multiple fields", "multiphysics" and "multi-components". After having read the paper carefully, it seems to me that the main difference between "multiphysics" and "multi-components" is in the way they are treated computationally. The multiple components being treated as interchangeable, in the sense that the same model is used for the conditioning of one on the others, while multiple "physics" do not assume such interchangeability.
- l253. The paper would benefit greatly by providing at this stage a clear examples of what a "component" is.

**Questions:**

- You mention several times that the model "learns the energy", as well as the conditional energies, but do you actually learn the energy function E? Or, do you learn the gradient of the energy function (that is the scores, and conditional scores), thanks to the denoiser?
- As your mention, your method seems computationally intensive compared to a FNO for example, not just because of the denoising steps but also because of the loop over the physical fields. Do you have a rough estimation of how it compares in terms of execution time and in terms of FLOPs?
- In algorithms 1 and 2, could you clarify what the "outer inputs" are in comparison to the "physical fields"?

---

> ### Author Response · Authors · 2024-11-22
> **Official Response to Reviewer 9cxV (1)**
>
> We thank the reviewer for the constructive review, and are glad that the reviewer recognizes the novelty of our method. In the following, we address the points the reviewer raised about, and answer the questions.
>
> >Re1: claim (2): There are several datasets in the literature that can be  described as simulations of multiphysics, multi-component, such as  PDEBench [Takamoto et al., 2022]. The authors should better precise why  the new datasets they propose is a contribution to the literature.
>
> Answer: Thanks for your comments. Multiphysics simulation refers to the simultaneous consideration and modeling of multiple **physical processes**, such as heat conduction, fluid flow, and structural mechanics, each of which may involve one or multiple physical fields. Therefore, multiphysics does not simply mean multiple physical fields, i.e., there are many scenarios that involve several physical fields, but only one physical process. In the "PDEBench" [1], although a series of partial differential equation (PDE) cases are provided, including convection, diffusion reaction, and Navier-Stokes equations, there are no cases of multiphysics simulation, such as coupling the reaction-diffusion equation with the Navier-Stokes equations, which is very common in combustion dynamics. In this manuscript, Experiment 1 is to calculate the reaction-diffusion equation, which does exist in the literature [1]. But strictly speaking, this does not constitute a multiphysics problem because they all belong to the concentration field; it is merely for validating the proposed algorithm. We explain this in the updated manuscript. Experiment 2 is a simplified version of the nuclear thermal coupling problem in the nuclear engineering field, encompassing three physical processes, with the fluid field solved using the finite volume method and the other two fields solved using the finite element method. This problem covers most coupling types: strong and weak coupling between physical processes, regional and interface coupling, and unidirectional and bidirectional coupling, making it a very representative multiphysics simulation case. To our knowledge, such cases **have not been found** in literature [1] or other literature.
>
> Multi-component simulation refers to the simulation of complex structures composed of multiple similar components, which is very common in civil, aerospace, and nuclear engineering fields. Multi-component simulation typically requires that in the inference time, it can due with **larger structures** than previously seen. For example, the reactor core typically consists of hundreds or thousands of fuel elements arranged in a square or hexagonal pattern. In fuel cells, the repeated array of ribs directly affects the cell's performance. The case domains in the PDEBench [1] are all on **regular** rectangular domains with relatively simple geometric structures and **no interaction** between multiple structural components. On the other hand, Experiment 3 in this manuscript solves the thermal and mechanical properties of multiple prismatic fuel elements, with mechanical and thermal interactions between adjacent fuel elements. Our predicted structure is more complex than in training, involving a large structure of 64 elements, each containing 804 grid points. Therefore, we consider this problem to be a typical case of multi-component problems with a certain level of complexity.
>
> We add appendix K and provide Table 21 to outline the principal characteristics of these datasets and compare them with Reference [1]. The number of physics processes and the number of components are all **1** in Reference [1]. The table is also provided in next responce.
>
> [1] Takamoto, Makoto, et al. "PDEBench: An extensive benchmark for scientific machine learning." Advances in Neural Information Processing Systems 35 (2022): 1596-1611.

---

> ### Author Response · Authors · 2024-11-22
> **Official Response to Reviewer 9cxV (2)**
>
> | PDE                                                                           | Nd  | Time | Computational domain | Number of physics processes | Number of Component |
> | ----------------------------------------------------------------------------- | --- | ---- | -------------------- | ------------------------ | ------------------- |
> | advection                                                                     | 1   | yes  | Line                 | 1                        | 1                   |
> | Burgers'                                                                      | 1   | yes  | Line                 | 1                        | 1                   |
> | reaction-diffusion                                                            | 1   | yes  | Line                 | 1                        | 1                   |
> | reaction-diffusion                                                            | 2   | yes  | Rectangle            | 1                        | 1                   |
> | diffusion-sorption                                                            | 1   | yes  | Line                 | 1                        | 1                   |
> | compressible Navier-Stokes                                                    | 1   | yes  | Line                 | 1                        | 1                   |
> | compressible Navier-Stokes                                                    | 2   | yes  | Rectangle            | 1                        | 1                   |
> | compressible Navier-Stokes                                                    | 3   | yes  | Cube                 | 1                        | 1                   |
> | incompressible Navier-Stokes                                                  | 2   | yes  | Rectangle            | 1                        | 1                   |
> | Darcy flow                                                                    | 2   | no   | Rectangle            | 1                        | 1                   |
> | shallow-water                                                                 | 2   | yes  | Rectangle            | 1                        | 1                   |
> | **reaction-diffusion (Exp1)**                                                 | 1   | yes  | Line                 | 1                        | 1                   |
> | **heat conduction + neutron diffusion + incompressible Navier-Stokes (Exp2)** | 2   | yes  | 3 Rectangle          | 3                        | 1                   |
> | **heat conduction + mechanics (Exp3)**                                        | 2   | no   | Irregular domain     | 2                        | 16 / 64             |
>
> Thus, we believe that our datasets in Exp2 and Exp3 constitute a contribution to the literature, which consists of multiphysics and multi-component simulations not present in previous benchmarks.
>
> >Re2: "a process we have mathematically proven", I don't think a  "process" qualifies as being able to be "mathematically proven". You  should precise what is "mathematically proven";
> "This reverse diffusion process is also mathematically validated", same remark
>
> Answer: Thank you for pointing out the inaccuracies in our previous statement. Indeed, a process should not be described as "be proven". What we want to express here is that we have mathematically derived the principles why our algorithm can obtain coupled solutions and large structure solutions. This has been corrected in the new manuscript.

---

> ### Author Response · Authors · 2024-11-22
> **Official Response to Reviewer 9cxV (3)**
>
> >Re3: "there do not exist utilized machine learning methods for  multi-component simulation". I don't really understand the novelty  since, technically, UNets [Ronneberger et al. 2015], FNO [Li et al.  2022], Transformer [Mccabe et al. 2023], diffusion [Kohl et al. 2024]  models are already used on multi-component simulations.
>
> Answer: In our responses to the above comments Re1, we have provided extensive explanations on the multi-component issue. In short, multi-component simulation refers to the simulation of complex structures composed of multiple similar components. Multi-component simulation typically requires that in the inference time, it can due with **larger structures** than previously seen. As was discussed in the response to the above Re1, previous benchmarks do not possess multi-component datasets, and the methods you mentioned were not specifically designed for multi-component simulations, yet some of them may be suitable baselines. Here, we provide a detailed discussion on the mentioned methods about their applicability to multi-component simulations:
> - For the  U-Net [1], although it has been applied to many learning physical simulation works before, it requires the data to form a regular grid. In contrast, multi-component problems typically have non-rectangular grids. Thus, U-Net is in general not applicable to multi-component problems.
> - For the FNO [2], due to its FFT and IFFT being global operations in the full domain, it requires that the training and inference have the same domain structure. In contrast, multi-component simulation typically requires to simulate larger structures than seen in training, which FNO cannot handle.
> - For the Transformer [3], it has the potential to address multi-component simulations, since it allows to handle larger input length during inference. The standard transformer [3,7] without graph structure embedding cannot handle multi-component simulation well, since they does not consider the complex graphical interactions between the components. On the other hand, graph transformers, such as SAN [5] which utilizes the eigenvectors as position embeddings, has the potential to address multi-component simulations, which we have added as a baseline in the revised manuscript.
> - For the diffusion [4], the three tasks addressed in this article are: Incompressible Wake Flow, Transonic Cylinder Flow, and Isotropic Turbulence. The first two tasks simulate the flow of fluid around a cylinder in a rectangular domain, while the third task involves turbulence within a cuboid, with the computational domain being a 2D slice from a 3D dataset. These tasks are not related to multi-component problems. However, if the simulation involves a complex phenomenon such as fluid flowing around a **bundle** of rods which is a multi-component problem, our method might be applicable for solving it.
>
> We have also modified the phrasing of this sentence in the Related Work to make it more accurate. We have also added the baselines of Graph Transformer SAN [5] and Graph Neural Network GIN [6] in Experiment 3 to compare the performance. Due to the uniformity of graph structures in all training data and the fact that SAN learns a global relationship, SAN fails to predict larger structures. We see that our method that is specifically designed for multi-component simulation, significantly outperforms the baselines. Part of Table 3 is also provided as follows (the unit is $1\times 10^{-2}$):
> | Method             | single T | single ε | 16-component T | 16-component ε | 64-component T | 64-component ε |
> | ------------------ | -------- | -------- | -------------- | -------------- | -------------- | -------------- |
> | GIN              |    -     |     -    | 1.96           | 3.18           | 4.63           | 7.02           |
> | SAN              |    -     |     -    | 0.114           | 16.5           | 100           | 11800           |
> | MultiSimDiff (ours) | 0.107    | 0.303    | 0.213          | 1.03           | 0.759          | 1.94           |
>
>
> [1]Ronneberger, et al. "U-net: Convolutional networks for biomedical image segmentation." 2015.
>
> [2]Wen, Gege, Z Li, et al. "U-FNO—An enhanced Fourier neural operator-based deep-learning model for multiphase flow." 2022.
>
> [3]Alwahas, Areej, Kasper Johansen, and Matthew McCabe. "Crop Type Mapping Using Self-supervised Transformer with Energy-based Graph Optimization in Data-Poor Regions." 2023.
>
> [4]Kohl, Georgi, et al. "Benchmarking autoregressive conditional diffusion models for turbulent flow simulation." 2024.
>
> [5] Kreuzer, Devin, et al. "Rethinking graph transformers with spectral attention." Advances in Neural Information Processing Systems 34 (2021): 21618-21629.
>
> [6] Keyulu Xu, Weihua Hu, Jure Leskovec, and Stefanie Jegelka. How powerful are graph neural networks? In International Conference on Learning Representations, 2019.
>
> [7] Vaswani, A. "Attention is all you need." Advances in Neural Information Processing Systems (2017).

---

> > ### Author Response · Authors · 2024-11-22
> > **Official Response to Reviewer 9cxV (4)**
> >
> > >Re4: Paragraph 3.2 is confusing. In particular, there can be  confusion between "multiple fields", "multiphysics" and  "multi-components". After having read the paper carefully, it seems to  me that the main difference between "multiphysics" and  "multi-components" is in the way they are treated computationally. The  multiple components being treated as interchangeable, in the sense that  the same model is used for the conditioning of one on the others, while  multiple "physics" do not assume such interchangeability.
> >
> > Answer: We have revised some of the content in Section 3.2 to enhance its clarity. In short, a multiphysics problem is composed of multiple physical processes, where each process may contain one or more fields. For example, the mechanics contains the stress and strain fields in three directions. Multi-component refers to multiple **entities**, and if there are multiple physical processes on this entity, it is also a multiphysics problem.
> >
> > >Re5: The paper would benefit greatly by providing at this stage a clear examples of what a "component" is.
> >
> > Answer: In the above answers, we have explained what multi-component is. We have further added a definition of component in Section 1 in the revised manuscript: a repeatable basic unit that makes up a complete structure. We also revised the Figure 1 in the manuscript to provide a clear example of a component and multi-component simulation.
> >
> > >Re6: You mention several times that the model "learns the energy", as  well as the conditional energies, but do you actually learn the energy  function E? Or, do you learn the gradient of the energy function (that  is the scores, and conditional scores), thanks to the denoiser?.
> >
> > Answer: Thanks for the question. Yes, we actually learn the gradient of the energy function through the diffusion model. We have updated the manuscript and now it clearly states that we learn the gradient of the energy.

---

> ### Author Response · Authors · 2024-11-22
> **Official Response to Reviewer 9cxV (5)**
>
> >Re7: As your mention, your method seems computationally intensive  compared to a FNO for example, not just because of the denoising steps  but also because of the loop over the physical fields. Do you have a  rough estimation of how it compares in terms of execution time and in  terms of FLOPs?
>
> Answer: First, we would like to clarify that our algorithm is applicable to various network architectures, including FNO and U-Net. Therefore, the high computational cost you mentioned likely refers to the greater computational cost of U-Net compared to FNO. For a specific network architecture, we have trained **both** diffusion models and surrogate models with identical network parameter settings. This means that the time complexity for a single prediction by the neural network is almost the **same** for both the surrogate model and the diffusion model. However, the main difference between the two lies in the fact that the diffusion model requires multiple denoising steps to reach the final solution, while the surrogate model needs to iterate with other physical processes or components until the solutions on each physical process or component converge. We have added a comparative analysis of time complexity and computational time in appendix I. Compared to numerical programs, our method achieves **29 times** and **41 times** acceleration in experiments 2 and 3, respectively. When compared to the surrogate model, our algorithm takes approximately 2-3 times longer to run for multiphysics problems. For multi-component problems, our algorithm is more efficient, with a running time of about half that of the surrogate model.
>
> Similarly, the number of floating-point operations (FLOPs) required for a single neural network prediction is almost identical for both the surrogate model and the diffusion model. However, due to the difficulty in calculating FLOPs in numerical procedures, we have only conducted a comparison of runtimes. Table 20 is also provided as follows, the time unit for each experiment is defined as the time required for a single neural network inference.
>
> | Experiment                            | Unit (s) | Numerical program | Surrogate model | model | Speedup |
> | ------------------------------------- | -------- | ----------------- | --------------- | ----- | ------- |
> | Exp 1: Reaction-diffusion                    | 0.0115   | 6                 | 54              | 100   | 0.064   |
> | Exp 2: Nuclear thermal coupling              | 0.0242   | 4368              | 63              | 150   | 29      |
> | Exp 3: Prismatic fuel element (16-component) | 0.0067   | 834               | 309             | 150   | 5.6     |
> | Exp 3: Prismatic fuel element (64-component) | 0.0256   | 6170              | 324             | 150   | 41      |
>
> >Re8: In algorithms 1 and 2, could you clarify what the "outer inputs" are in comparison to the "physical fields"?
>
> Answer: Thank you for pointing out this issue. We are sorry that we didn't provide a clear definition before, but now we have updated it in the form of a footnote 1 in the manuscript. In this manuscript, "outer inputs" refers to the inputs of the physical system, physical fields are what we need to solve. For example, in experiment 1, they are the initial conditions; in experiment 2, they are the variations of boundary neutron flux density over time; and in experiment 3, they are the heat flux density.

---

> > ### Author Response · Authors · 2024-11-25
> > **A gentle reminder: please respond to our rebuttal**
> >
> > Dear Reviewer 9cxV,
> >
> > Thank you for your time and effort in reviewing our work. We have carefully considered your detailed comments and questions, and we have tried to address all your concerns accordingly.
> >
> > As the deadline for author-reviewer discussions is approaching, could you please go over our responses? If you find our responses satisfactory, we hope you could consider adjusting your initial rating. Please feel free to share any additional comments you may have.
> >
> > Thank you!
> >
> > Authors

---

> > > ### Comment · Area_Chair_N5rq · 2024-11-26
> > >
> > > Dear reviewer,
> > >
> > > Please make to sure to read, at least acknowledge, and possibly further discuss the authors' responses to your comments. Update or maintain your score as you see fit.
> > >
> > > The AC.

---

> > > ### Comment · Reviewer_9cxV · 2024-12-02
> > >
> > > I thank the authors for their detailed response to my points, which helped clarify important aspects of the paper. I will raise my score to reflect this.
> > >
> > > However, I once again encourage the authors to formalize what is meant by "multi-component" and "multi-physics".
> > > In your response, you write, "Multiphysics simulation refers to the simultaneous consideration and modeling of multiple physical processes", which is clear. But, how does this differ from coupled PDEs? While the reader might infer the distinction, it should be explicitly stated to better motivate the choice of baselines.
> > > More importantly, you define "Multi-component simulation" as "the simulation of complex structures composed of multiple similar components". This definition is inadequate because it does not clearly explain what a "component" entails.
> > >
> > > Clarifying these is essential for understanding the task presented in the paper and for justifying the choice of baselines and the paper's contributions.

---

> > > > ### Author Response · Authors · 2024-12-02
> > > > **Official Comment by Authors**
> > > >
> > > > Thank you for your constructive comments and for raising our manuscript's score. Your insights have been invaluable in enhancing the quality of our work.
> > > >
> > > > Regarding the multiphysics, your understanding is correct. Indeed, in mathematics, multiphysics problems are represented by a set of coupled partial differential equations (PDEs) that describe multiple physical processes. These coupled PDEs, which describe a single physical process, are generally solved together. However, in the engineering field, coupled PDEs that describe multiple physical processes are typically solved separately through data transfer methods due to factors such as the difference in spatial and temporal resolution (as mentioned in lines 76-78 of the manuscript). For instance, in reactor engineering, the nuclear-thermal coupling involves solving both the neutron physics field and the fluid field. The fluid field is addressed using the Navier-Stokes (NS) equations, which include multiple PDEs to describe the fluid's flow and heat transfer phenomena, and are typically solved using the finite volume method on a fine mesh. The neutron physics field is addressed using a set of diffusion equations, which are generally solved using finite difference methods or Monte Carlo techniques on a very coarse mesh. Due to the different solution methods and spatial-temporal resolutions, these two fields can only be solved separately. Previous studies on coupled PDEs focused on a single physical process, whereas Experiment II in this paper addresses three physical processes.
> > > >
> > > > Regarding the multi-component, the term "component" has been defined in the manuscript at line 39 as follows: "Component is defined as: a repeatable basic unit that makes up a complete structure." We also give a further explanation at the beginning of Section 3.2.
> > > >
> > > > We will incoporate the above discussion in the manuscript.

---

### Official Review · Reviewer_xu7A · 2024-11-04

**Soundness:** 2
**Presentation:** 3
**Contribution:** 2
**Rating:** 6
**Confidence:** 4

**Summary:**

This paper presents a data-driven approach for multiphysics and multi-component simulations using compositional generative diffusion models. The proposed method tackles the complexities of coupled simulations by learning energy functions that model conditional probabilities between physical fields and components. The proposed method is validated on 3 tasks, including reaction-diffusion, nuclear thermal coupling, and thermal-mechanical simulations of prismatic fuel elements. The results show improved accuracy and reduced error over traditional surrogate models.

**Strengths:**

1. The use of compositional generative models for multiphysics and multi-component simulations is a fresh approach in this field, addressing the challenges of coupling multiple physical domains.
2. This paper demonstrates the capability of MultiSimDiff to predict coupled interactions from models trained on decoupled data, simplifying data requirements and model development.

**Weaknesses:**

1. Only accuracy is compared in the examples. Efficiency comparison is also important. Specifically, a fair comparison with standard numerical methods (such as FEM or FVM) is important.
2. The iterative nature of the diffusion process in MultiSimDiff can be computationally intensive, particularly for multiphysics simulations due to the high complexity.
3. Since the model is trained on decoupled data, the approach's success may depend on the quality of this initial data. The paper could benefit from further discussion on the robustness of MultiSimDiff when the decoupled data does not closely resemble the coupled dynamics.

**Questions:**

1. There is only one example for demonstrating training one small structure simulation data and predicting larger structures. Graph Neural Network also has similar abilities. How's the performance comparison?
2. How does MultiSimDiff handle cases where the decoupled training data does not closely match the dynamics of coupled data? Would the model performance degrade significantly?
3. Could you clarify if there are specific scenarios where traditional numerical solvers might still outperform MultiSimDiff in terms of accuracy or computational cost?

---

> ### Author Response · Authors · 2024-11-22
> **Official Response to Reviewer xu7A (1)**
>
> We thank the reviewer for the constructive review, and are glad that the reviewer recognizes the novelty and effectiveness of our method. In the following, we address the points the reviewer raised about, and answer the questions.
>
>
> >Re1: Only accuracy is compared in the examples. Efficiency comparison is  also important. Specifically, a fair comparison with standard numerical  methods (such as FEM or FVM) is important.
>
> Answer: Thank you for the suggestion. We have added the efficiency comparison in appendix I in the revised manuscript, the results are shown in Table 20 and also below. The time unit for each experiment is defined as the time required for a single neural network inference. The last column shows our method's speedup compared to the numerical program.
>
> | Experiment                            | Unit (s) | Numerical program | Surrogate model | model | Speedup |
> | ------------------------------------- | -------- | ----------------- | --------------- | ----- | ------- |
> | Exp 1: Reaction-diffusion                    | 0.0115   | 6                 | 54              | 100   | 0.064   |
> | Exp 2: Nuclear thermal coupling              | 0.0242   | 4368              | 63              | 150   | 29      |
> | Exp 3: Prismatic fuel element (16-component) | 0.0067   | 834               | 309             | 150   | 5.6     |
> | Exp 3: Prismatic fuel element (64-component) | 0.0256   | 6170              | 324             | 150   | 41      |
>
> From the table, we see that because Experiment 1 is simple, the numerical program runs very quickly and does not achieve acceleration. However, Experiment 2 and Experiment 3 achieve **accelerations of 29 and 41 times**, respectively. The more complex the problem, the higher the efficiency of our algorithm compared to numerical programs.
>
> >Re2: The iterative nature of the diffusion process in MultiSimDiff can be  computationally intensive, particularly for multiphysics simulations  due to the high complexity.
>
> Answer: We have employed DDIM for accelerated sampling, and the results are shown in appendix H. We also provide part of the Tables below. We see that compared to the original DDPM, DDIM sampling can achieve a **10-fold** and **5-fold** acceleration for multi-physics and multi-component problems, respectively, while maintaining accuracy. For DDPM, the diffusion step of all experiments is 250.
>
>
> | Exp1          | u (decoupled) | u (coupled) | v (decoupled) | v (coupled) | runtime|
> | ------------- | ------------- | ----------- | ------------- | ----------- | ----------- |
> | Original DDPM | 0.0119        | 0.0141      | 0.0046        | 0.0174      | 11.5 s |
> | $S=25$        | 0.0119        | 0.0151      | 0.0081        | 0.0192      | 1.15s  |
>
> | Exp2          | Neutron (decoupled) | Neutron (coupled) | Solid (decoupled) | Solid (coupled) | Fluid (decoupled) | Fluid (coupled) | runtime |
> | ------------- | ------------------- | ----------------- | ----------------- | --------------- | ----------------- | --------------- | ----------- |
> | Original DDPM | 0.487               | 1.97              | 0.108             | 2.87            | 0.303             | 3.91            | 36.3 s |
> | $S=25$        | 0.552               | 2.03              | 0.142             | 3.64            | 0.343             | 4.08            | 3.63s  |
>
> | Exp3          | Single T | Single ε | 16-component T | 16-component ε | 64-component T | 64-component ε | runtime (64-component) |
> | ------------- | -------- | -------- | -------------- | -------------- | -------------- | -------------- | -------------- |
> | Original DDPM | 0.107    | 0.303    | 0.213          | 1.03           | 0.759          | 1.94           | 19.2 s  |
> | $S=50$        | 0.158    | 0.337    | 0.669          | 1.87           | 0.865          | 2.31           | 3.84 s  |

---

> > ### Author Response · Authors · 2024-11-22
> > **Official Response to Reviewer xu7A (2)**
> >
> > >Re3: Since the model is trained on decoupled data, the approach's success  may depend on the quality of this initial data. The paper could benefit  from further discussion on the robustness of MultiSimDiff when the  decoupled data does not closely resemble the coupled dynamics.
> >
> > Answer: Thank you for raising this important point. Actually, our **decoupled data differs significantly from the coupled dynamics**. To illustrate this, we add a comparison between the training and testing datasets to highlight their significant differences in appendix G in the revised manuscript. We calculate the Wasserstein distances between decoupled and coupled data in multiphysics problems, as well as between small and large structural data in multi-component problems, which are compared with the Wasserstein distances between training and validation data. In addition, we also used the t-SNE algorithm to visualize this difference in Figures 11, 12, 13 in the revised manuscript. Table 15 below indicates that there is a **significant difference** between the decoupled data used for training and the coupled data used for testing. Therefore, even if the data for training data does not closely resemble the coupled dynamics, our algorithm can correctly predict the coupled solution. Therefore, our MultiSimDiff is **robust** when the decoupled data differs significantly from the coupled dynamics.
> > |                          | Training and validation | Training and testing |
> > | ------------------------ | ----------------------- | -------------------- |
> > | Reaction-diffusion       |                         |                      |
> > | $u$                      | 0.343                   | 52.7                 |
> > | $v$                      | 0.0435                  | 20.3                 |
> > | Nuclear thermal coupling |                         |                      |
> > | neutron                  | 42.4                    | 31.3                 |
> > | solid                    | 1.35                    | 56.3                 |
> > | fluid                    | 1.22                    | 986                  |
> > | Prismatic fuel element   |                         |                      |
> > | $T$                      | 0.233                   | 9.05                 |
> > | $\varepsilon$            | 0.625                   | 12.5                 |
> >
> > >Re4: There is only one example for demonstrating training one small  structure simulation data and predicting larger structures. Graph Neural  Network also has similar abilities. How's the performance comparison?
> >
> > Answer: Thank you for suggesting the baseline. We have incorporated the Graph Isomorphism Network (GIN) [1] and the Spectral Attention Network (SAN) [2], which were suggested by Reviewer 9cxV, as robust benchmarks for graph neural networks (GNNs) and graph transformers, respectively. We also conduct hyperparameter search to obtain the best performance. GIN and SAN are trained on small graphs with 16 components and tested on large graphs with 64 components. Due to the uniformity of graph structures in all training data (i.e.) and the fact that SAN learns a global relationship, SAN fails to generalize to larger structures. We have updated Table 3, the results indicate that our method has significantly better performance. Part of Table 3 is also provided as follows (the unit is $1\times 10^{-2}$):
> > | Method             | single T | single ε | 16-component T | 16-component ε | 64-component T | 64-component ε |
> > | ------------------ | -------- | -------- | -------------- | -------------- | -------------- | -------------- |
> > | GIN              |    -     |     -    | 1.96           | 3.18           | 4.63           | 7.02           |
> > | SAN              |    -     |     -    | 0.114           | 16.5           | 100           | 11800           |
> > | MultiSimDiff (ours) | 0.107    | 0.303    | 0.213          | 1.03           | 0.759          | 1.94           |
> >
> >
> > [1] Keyulu Xu, Weihua Hu, Jure Leskovec, and Stefanie Jegelka. How powerful are graph neural networks? In International Conference on Learning Representations, 2019.
> >
> > [2] Kreuzer, Devin, et al. "Rethinking graph transformers with spectral attention." Advances in Neural Information Processing Systems 34 (2021): 21618-21629.

---

> ### Author Response · Authors · 2024-11-22
> **Official Response to Reviewer xu7A (3)**
>
> >Re5: How does MultiSimDiff handle cases where the decoupled training data  does not closely match the dynamics of coupled data? Would the model  performance degrade significantly?
>
> Answer: In fact, the training data used in experiments 1 and 2 does not closely resemble the coupled dynamics, but our algorithm can correctly predict the coupled solution. Details can be found in the above response to Re3 and the appendix G in the revised manuscript.
>
> >Re6: Could you clarify if there are specific scenarios where traditional  numerical solvers might still outperform MultiSimDiff in terms of  accuracy or computational cost?
>
> Answer: Firstly, in terms of accuracy, since we currently regard the numerical program's solution as the ground truth, the accuracy can only be as close as possible to the numerical program. Regarding efficiency, for very simple problems, such as the solution of the reaction-diffusion equation in experiment 1, our algorithm does not hold an advantage. However, for slightly more complex cases, like experiments 2 and 3, our algorithm is capable of achieving significant acceleration. The more complex the problem, the higher the efficiency of our algorithm compared to numerical programs.

---

> > ### Author Response · Authors · 2024-11-25
> > **A gentle reminder: please respond to our rebuttal**
> >
> > A gentle reminder: please respond to our rebuttal
> > Dear Reviewer xu7A,
> >
> > Thank you for your time and effort in reviewing our work. We have carefully considered your detailed comments and questions, and we have tried to address all your concerns accordingly.
> >
> > As the deadline for author-reviewer discussions is approaching, could you please go over our responses? If you find our responses satisfactory, we hope you could consider adjusting your initial rating. Please feel free to share any additional comments you may have.
> >
> > Thank you!
> >
> > Authors

---

> > > ### Comment · Area_Chair_N5rq · 2024-11-26
> > >
> > > Dear reviewer,
> > >
> > > Please make to sure to read, at least acknowledge, and possibly further discuss the authors' responses to your comments. Update or maintain your score as you see fit.
> > >
> > > The AC.

---

> ### Author Response · Authors · 2024-12-03
> **A gentle reminder: please respond to our rebuttal Dear Reviewer xu7A**
>
> Thank you for your time and effort in reviewing our work. We have carefully considered your detailed comments and questions, and we have tried to address all your concerns accordingly. As the deadline for author-reviewer discussions is approaching, could you please go over our responses? If you find our responses satisfactory, we hope you could consider adjusting your initial rating. Please feel free to share any additional comments you may have.
>
> Thank you!
>
> Authors

---

### Author Response · Authors · 2024-11-22
**General Response**

We thank the reviewers for their thorough and constructive comments. We are glad that the reviewers recognize that our method is novel (xu7a, 9cxv,HRRS,FSCM,1Pcu), our experimental results are convincing (xu7a,FSCM), our dataset is practical (1Pcu,FSCM).

Based on the reviewers' valuable feedback, we have conducted a number of additional experiments and revised the manuscript, which hopefully resolves the reviewers' concerns. The major additional experiments and improvements are as follows:

1. We add a comparison between the training and testing datasets to highlight their significant differences, as raised by Reviewer xu7A. We calculate Wasserstein distances and use the t-SNE algorithm to visualize this difference in Figures 11, 12, and 13. The results and analysis are in Appendix G. This significant difference between training and testing datasets illustrates the difficulty of the task and also demonstrates the capabilities of our algorithm. For more details, please see the responses to Reviewer xu7A.
3. We use DDIM to accelerate sampling and compare the run time of numerical programs, surrogate models, and our method, as raised by Reviewer xu7A, 9cxV, HRRS, 1Pcu. The results and analysis of DDIM sampling are in Appendix H. The comparison of the running time is in Appendix I. We find that our method is efficient and achieves accelerations of 29 and 41 times compared with numerical programs in experiment 1 and experiment 2, respectively.
4. We add the application scenarios of our algorithm for multi-component simulation, as suggested by Review 1Pcu. The analysis is in Appendix J. Through analysis, we have gained a deeper understanding of the applicability and reasons behind our algorithm. For more details, see the responses to Reviewer 1Pcu.
5. We add two new baselines for multi-component simulation: Graph neural network and Graph transformer, as suggested by Reviewer xu7A, 9cxV. Our algorithm significantly outperforms the baselines, demonstrating its capability for modeling multi-component problems and generalizing to more complex structures. For more details, see the responses to Reviewer xu7A, 9cxV.
6. We add an overall description of our datasets and compare them with existed datasets, as suggested by Reviewer 9cxV. The datasets of experiments 2 and 3 are a contribution to the community which offer multiphysics and multi-component aspects. For more details, see the responses to Reviewer 9cxV.
7. We have updated some of the expressions in the manuscript to make it easier for the readers to understand what multiphysics and multi-component simulation are focused on in this manuscript, as raised by Reviewer 9cxV. Multiphysics consists of multiple physical processes, where each process may contain one or more fields. Multi-component is a complex structure composed of multiple similar components. Component is defined as: a repeatable basic unit that makes up a complete structure. Multi-component simulation typically requires to generalize to larger structures than in training. The solution on components can also involve multiple physical processes.
8. We add an explanation for why other compositional models are not suitable for our tasks, as raised by Reviewer 1Pcu.

---

> ### Author Response · Authors · 2024-12-02
> **A gentle reminder: please respond to our rebuttal Dear Reviewers**
>
> Thank you for your time and effort in reviewing our work. We have carefully considered your detailed comments and questions, and we have tried to address all your concerns accordingly.
> As the deadline for author-reviewer discussions is approaching, could you please go over our responses? If you find our responses satisfactory, we hope you could consider adjusting your initial rating. Please feel free to share any additional comments you may have.
>
> Thank you!
>
> Authors

---

### Comment · Area_Chair_N5rq · 2024-11-26

Dear all,

The deadline for the authors-reviewers phase is approaching (December 2).

@For reviewers, please read, acknowledge and possibly further discuss the authors' responses to your comments. While decisions do not need to be made at this stage, please make sure to reevaluate your score in light of the authors' responses and of the discussion.

- You can increase your score if you feel that the authors have addressed your concerns and the paper is now stronger.
- You can decrease your score if you have new concerns that have not been addressed by the authors.
- You can keep your score if you feel that the authors have not addressed your concerns or that remaining concerns are critical.

Importantly, you are not expected to update your score. Nevertheless, to reach fair and informed decisions, you should make sure that your score reflects the quality of the paper as you see it now. Your review (either positive or negative) should be based on factual arguments rather than opinions. In particular, if the authors have successfully answered most of your initial concerns, your score should reflect this, as it otherwise means that your initial score was not entirely grounded by the arguments you provided in your review. Ponder whether the paper makes valuable scientific contributions from which the ICLR community could benefit, over subjective preferences or unreasonable expectations.

@For authors, please respond to remaining concerns and questions raised by the reviewers. Make sure to provide short and clear answers. If needed, you can also update the PDF of the paper to reflect changes in the text. Please note however that reviewers are not expected to re-review the paper, so your response should ideally be self-contained.

The AC.

---

### Meta-Review · Area_Chair_N5rq · 2024-12-21

**Metareview:**

The reviewers are divided (6-5-5-8-5-5) about the paper, leaning more towards rejection than acceptance. The paper presents a data-driven approach for multiphysics and multi-component simulations using compositional generative diffusion models. Concerns have been raised regarding the presentation of the method. The author-reviewer discussion has been constructive and has led to a number of clarifications, but some of the reviewers' concerns remain. Regarding the experimental validation, concerns were raised regarding the computational efficiency of MultiSimDiff and its robustness to coupled interactions. The authors have provided new results using DDIM for accelerated sampling, which show a significant speedup, and have provided further results exploring the case of coupled data. Finally, the novelty of the approach has been questioned, but the authors have provided clarifications. Unfortunately, some of the reviewers have not replied back to the authors' clarifications. Given the lack of a strong signal towards acceptance, the remaining concerns, and the mixed reviews, I recommend rejection. I encourage the authors to address the reviewers' comments and to resubmit to a future conference.

**Additional Comments On Reviewer Discussion:**

The author-reviewer discussion has been constructive and has led to a number of clarifications, but some of the reviewers' concerns remain.

---

### Decision · Program_Chairs · 2025-01-22

Reject